# Impact of inner ear malformation and cochlear nerve deficiency on the development of auditory-language network in children with profound sensorineural hearing loss

Yaoxuan Wang[1,2,3†], Mengda Jiang[4†], Yuting Zhu[1,2,3†], Lu Xue[1,2,3], Wenying Shu[1,2,3], Xiang Li[1,2,3], Hongsai Chen[1,2,3], Yun Li[1,2,3], Ying Chen[1,2,3], Yongchuan Chai[1,2,3], Yu Zhang[1,2,3], Yinghua Chu[5], Yang Song[6], Xiaofeng Tao[4], Zhaoyan Wang[1,2,3]*, Hao Wu[1,2,3]*

[1]Department of Otolaryngology, Head and Neck Surgery, Shanghai Ninth People's Hospital, Shanghai Jiao Tong University School of Medicine, Shanghai, China; [2]Ear Institute, Shanghai Jiao Tong University School of Medicine, Shanghai, China; [3]Shanghai Key Laboratory of Translational Medicine on Ear and Nose diseases, Shanghai, China; [4]Department of Radiology, Shanghai Ninth People's Hospital, Shanghai Jiao Tong University School of Medicine, Shanghai, China; [5]MR Collaboration, Siemens Healthineers Ltd, Shanghai, China; [6]MR Scientific Marketing, Siemens Healthineers Ltd, Shanghai, China

*For correspondence:
wzyent2019@163.com (ZW);
wuhao@shsmu.edu.cn (HW)

†These authors contributed equally to this work

**Abstract** Profound congenital sensorineural hearing loss (SNHL) prevents children from developing spoken language. Cochlear implantation and auditory brainstem implantation can provide partial hearing sensation, but language development outcomes can vary, particularly for patients with inner ear malformations and/or cochlear nerve deficiency (IEM&CND). Currently, the peripheral auditory structure is evaluated through visual inspection of clinical imaging, but this method is insufficient for surgical planning and prognosis. The central auditory pathway is also challenging to examine in vivo due to its delicate subcortical structures. Previous attempts to locate subcortical auditory nuclei using fMRI responses to sounds are not applicable to patients with profound hearing loss as no auditory brainstem responses can be detected in these individuals, making it impossible to capture corresponding blood oxygen signals in fMRI. In this study, we developed a new pipeline for mapping the auditory pathway using structural and diffusional MRI. We used a fixel-based approach to investigate the structural development of the auditory-language network for profound SNHL children with normal peripheral structure and those with IEM&CND under 6 years old. Our findings indicate that the language pathway is more sensitive to peripheral auditory condition than the central auditory pathway, highlighting the importance of early intervention for profound SNHL children to provide timely speech inputs. We also propose a comprehensive pre-surgical evaluation extending from the cochlea to the auditory-language network, showing significant correlations between age, gender, Cn.VIII median contrast value, and the language network with post-implant qualitative outcomes.

## Editor's evaluation

This important study used high-resolution brain imaging methods to visualize and index non-invasively auditory and language pathways of young children born with inner ear malformations or cochlear nerve dysfunction resulting in profound hearing loss. Nerve fibre impairments were compellingly demonstrated in subcortical auditory and cortical language pathways relative to typical hearing controls. Qualitative language assessment and audiometry linked these structural findings with functional outcomes. The results suggested novel approaches for clinical assessment of central auditory and language pathways that may influence different intervention strategies.

## Introduction

Congenital profound hearing loss (hearing level >90 dB diagnosed at birth or in early childhood) deprives children of spoken language development and has lifelong negative consequences for education, employment, and psychosocial status (*Kral and O'Donoghue, 2010*). The prevalence of neonatal hearing loss has increased from 1.09 to 1.7 cases per 1000 live births in the past two decades (*CDC Early Hearing Detection and Intervention (EHDI) Hearing Screening & Follow-up Survey (HSFS), 2021*; *Alicia and Marcus, 2010*). Most congenital hearing loss is caused by sensorineural impairments in the inner ear, cochlear nerve, and/or central auditory pathway (*Korver et al., 2017*). The aetiology of these impairments is multifaceted, with genetic factors playing a significant role, alongside external contributors such as congenital infections (*Marazita et al., 1993*). Hearing loss is categorized as syndromic, with accompanying physical or laboratory findings, or non-syndromic, where hearing loss is the sole symptom and has a highly heterogeneous genetic basis (*Korver et al., 2017*). This article primarily focuses on non-syndromic sensorineural hearing loss (SNHL).

Inner ear malformations and cochlear nerve deficiencies (IEM&CND), identifiable with CT and MRI, contribute to 15–39% of paediatric SNHL cases (*Li et al., 2011*; *Mafong et al., 2002*). The rest is primarily due to cellular-level abnormalities. The aetiology of IEM&CND is complex and largely unknown, yet these abnormalities suggest an earlier developmental arrest compared to cases with normal inner ear structures (*Sennaroglu and Saatci, 2002*). Addressing IEM&CND is vital as it presents specific challenges in clinical management. Cochlear implantation (CI) and auditory brainstem implantation (ABI) are currently the only solutions for profound SNHL, but the choice between CI and ABI presents a dilemma for many IEM&CND patients. CI directly stimulates the spiral ganglion cells, the first-order neurons of the auditory pathway, while ABI bypasses the cochlear nerve and stimulates the second-order auditory neurons in the cochlear nucleus (CN) when complex IEM or cochlear nerve aplasia makes CI inapplicable (*Chen and Oghalai, 2016*). Both CI and ABI are capable of providing hearing sensation and language development for children with severe-to-profound prelingual hearing loss, but postoperative outcomes vary among individuals. ABI recipients generally have poorer speech recognition performance and delayed and incomplete language development compared to CI recipients (*Sennaroğlu et al., 2016*). However, the prognosis of CI may not necessarily be better than that of ABI for certain IEMs, including common cavity, cochlear hypoplasia, incomplete partition-type I, and cochlear aperture abnormalities, as the presence of sufficient cochlear fibres required for CI success is uncertain (*Freeman and Sennaroglu, 2018*; *Sennaroğlu and Bajin, 2017*). This uncertainty is hard to address for two reasons: (1) assessing the cochlear nerve through visual inspection of MRI poses challenges, including subjective limitations, image quality, and difficulty distinguishing the cochlear nerve from the cranial nerve VIII (the cochleovestibular nerve) in cases of common cavity; and (2) the absence of the cochlear nerve structurally does not always indicate a lack of functional hearing (*Thai-Van et al., 2000*). Furthermore, distinguishing certain types of IEMs can be problematic and may impact surgical decision-making. For example, differentiating cochlear aplasia with a dilated vestibule (CADV) from a common cavity is challenging; CADV is a definitive indication for ABI, while a common cavity allows for either CI or ABI. These observations underscore the importance of a deep dive into IEM&CND and highlight the limitations of contemporary clinical imaging in diagnosis, prognosis evaluation, and surgical guidance.

Speech is transmitted through the auditory pathway and understood at the language network (*Friederici, 2011*). For children with severe-to-profound prelingual hearing loss, the deprivation of auditory inputs into this workflow may affect the structural and functional development of associated brain regions, the degrees of impairment of which may be relevant to later auditory and language performance after surgical reconstruction of hearing. Diffusion tensor imaging (DTI) studies have

shown that children with profound SNHL exhibit decreased fractional anisotropy (FA) values along almost the entire auditory pathway, particularly in the inferior colliculus (IC), medial geniculate body (MGB), and auditory radiation (*Tarabichi et al., 2018*). Additionally, hearing and language outcomes (categories of auditory performance) 6–12 mo following CI were positively correlated with preoperative auditory pathway FA values (*Chang et al., 2012*; *Huang et al., 2015*; *Wang et al., 2019b*; *Wu et al., 2016*). Similarly, brain structures associated with the language pathway, including the superior temporal gyrus (STG), Broca's area, superior longitudinal fasciculus (SLF), and uncinate fasciculus (UF), also showed decreased FA values in children with profound hearing loss (*Wang et al., 2019b*; *Wang et al., 2019a*; *Wu et al., 2016*). The FA value of Broca's area was positively correlated with speech recognition performance after CI implantation (*Chang et al., 2012*). These results indicate that children with profound SNHL have a delayed or impaired microstructural organization in the central auditory pathway and the language pathway that may affect subsequent implantation outcomes.

However, there are several challenges in this field. Firstly, the human auditory pathway is difficult to inspect non-invasively due to its delicate nodes connected by curving and crossing fibres, especially the parts that are buried deep in the brainstem (*Zanin et al., 2019*). Functional MRI (fMRI) responses to natural sounds have been used to locate subcortical auditory structures (*Sitek et al., 2019*), but this method is not applicable to patients with profound hearing loss as no auditory brainstem responses (ABRs) can be detected in these individuals, making it impossible to capture corresponding blood oxygen signals in fMRI. In fact, few studies have located the auditory brainstem of children with profound hearing loss, particularly the CN that ABI targets. Therefore, new localization and tractography methods are needed to precisely map the auditory pathway in vivo for individuals with normal or impaired hearing. Secondly, although earlier studies reported several altered fibre tracts related to language function (*Chang et al., 2012*; *Wang et al., 2019b*; *Wang et al., 2019a*; *Wu et al., 2016*), there has been insufficient investigation into the language pathway. Its component streams, which are segmented on both structural and functional bases, do not correspond to the major fibre tracts typically studied in whole-brain analyses. Finally, the diffusion tensor model used in these studies performs poorly in estimating regions with crossing fibres (*Farquharson et al., 2013*), which may lead to problematic tractography (particularly in the auditory pathway) and false interpretations of FA alterations in children with profound hearing loss. To overcome this methodological limitation, the present study implemented a state-of-the-art fixel-based analysis (FBA) approach (*Dhollander et al., 2021*). A 'fixel' refers to an individual fibre population within a voxel, allowing for the quantification of white matter properties in fibre-crossing areas.

Furthermore, previous neuroimaging studies of profound congenital SNHL excluded children with IEM&CND. However, as mentioned earlier, these subjects comprise a significant proportion of patients with congenital hearing loss and are more inclined to be faced with difficult surgical decisions and unsatisfactory post-implantation outcomes. Therefore, it is important to include and focus on children with IEM&CND when studying central adaptations associated with profound hearing loss.

In the present study, we introduced a new pipeline for reconstructing the human auditory pathway and examined the brain structural development of children with profound congenital SNHL at both the acoustic processing level and the speech perception level. Specifically, we included children under the age of six with profound hearing loss, with and without IEM&CND, as well as normal hearing controls. We segmented the subcortical auditory nuclei using super-resolution track density imaging (TDI) maps and T1-weighted images, and tracked the auditory pathway and the language pathway using probabilistic tractography. Then, we used FBA to investigate the fibre properties of these two pathways. We aimed to investigate (1) the alteration of fibre metrics in the auditory-language network of children with profound SNHL; (2) the potential impact of IEM&CND on the pre-implant structural development of this network; and (3) the relationship between the pre-implant structural development of this network and the auditory-language outcomes following CI or ABI implantation. By doing so, we hope to provide new insights into the surgical strategies and rehabilitation of children with profound congenital SNHL.

**Table 1.** Demographic information for patients with congenital bilateral profound sensorineural hearing loss.

| Gender | Age (mo) | Gestational weeks | Birth weight (kg) | Inner ear structure (CT) | Cochlear nerve (CISS) | Surgery |
|---|---|---|---|---|---|---|
| M | 26 | 39 | 3.25 | Cochlear aplasia | Deficiency | ABI |
| M | 24 | 39 | 2.7 | Normal | Normal | CI |
| F | 56 | 39 | 3.2 | Normal | Normal | CI |
| F | 65 | 39 | 2.9 | Normal | Normal | CI |
| M | 17 | 40 | 3.4 | Incomplete partition type I | Deficiency | ABI candidate |
| M | 32 | 40 | 3.3 | Normal | Normal | CI |
| M | 29 | 40 | 3 | Cochlear hypoplasia (L), Cochlear aplasia (R) | Deficiency | ABI |
| M | 14 | 40 | 2.9 | Normal | Normal | CI |
| M | 6 | 39 | 4.3 | Hypoplastic cochlear aperture | Deficiency | ABI candidate |
| F | 72 | 36 | 2.2 | Hypoplastic cochlear aperture | Deficiency | ABI candidate |
| M | 44 | 40 | 3.97 | Normal | Normal | CI |
| F | 9 | 40 | 3.6 | Normal | Normal | CI |
| M | 8 | 38 | 6 | Hypoplastic cochlear aperture | Deficiency | ABI candidate |

ABI, auditory brainstem implantation; CI, cochlear implantation; CISS, constructive interference in steady state.

## Results

### Demographics

Twenty-three children aged under 6 years old including 13 patients with bilateral profound congenital SNHL (mean [SD] of age, 30.92 [6.115] mo; nine males) and 10 normal hearing volunteers (mean [SD] of age, 42.90 [4.270] months; five males) matched on age and gender were included (Mann–Whitney $U$ test: p-value for age = 0.077, 95% confidence interval for the difference in age = [–32.0, 6.5] mo; p-value for gender = 0.446, 95% confidence interval for the difference in male counts = [–1, 4]).

The audiological profiles of these patients, including newborn hearing testing results, initial diagnoses, interventions, and recent preoperative audiometric examination results, are presented in detail in *Supplementary file 1*. Nearly half of the patients (6 out of 13) had IEMs and cochlear nerve deficiencies: two of them underwent ABI surgeries; the other four were ABI candidates. The remaining half exhibited normal structures and underwent CI surgeries (see *Table 1*).

As for genetic testing, we obtained data for 7 of the 13 patients. Two patients had mutations in the GJB2 gene, one had a mutation in the OTOF gene, and one had a mutation in the MYO15A gene. Please refer to *Supplementary file 2* for detailed information on the genetic data.

Inner ear structure and cochlear nerve were inspected using temporal bone high-resolution CT and constructive interference in steady state (CISS) MRI. The presence of IEM&CND was determined based on Sennaroğlu classification criteria (*Sennaroğlu and Bajin, 2017*); descriptions refer to a bilateral condition if the side (such as L or R) is not specified.

### Segmentation of subcortical auditory regions and tractography of the central auditory pathway

All subcortical auditory nuclei, including the bilateral CN, superior olivary complex (SOC), IC, and MGB, were segmented in both group-average space and individual space with good inter-rater reliability (see *Figure 1*).

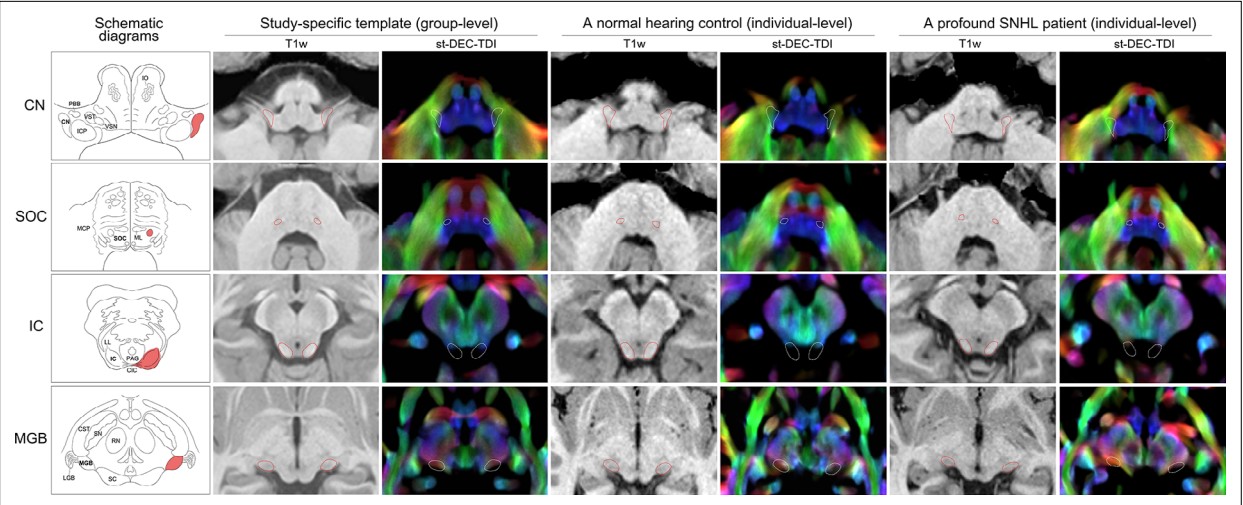

**Figure 1.** Segmentation of the subcortical auditory regions at group and individual levels. Subcortical auditory regions are exhibited as red areas in the schematic diagrams and dotted lines in T1w images and st-DEC-TDI maps. All images are in the axial plane. Schematic diagrams for the CN, SOC, and IC were redrawn from *Moore, 1987* and for the MGB from *Duvernoys Atlas of the Human Brain Stem and Cerebellum (2009)*. Group-level segmentation was performed in the study-specific template averaged from all 10 normal-hearing controls. Images from one normal hearing control (male, 32 months old) and one profound SNHL patient (male, 32 months old) were selected to show individual-level segmentation. CN, cochlear nucleus; ICP, inferior cerebellar peduncle; PBB, ponto bulbar body; VSN, trigeminal spinal nucleus; VST, trigeminal spinal tract; IO, inferior olive; SOC, superior olivary complex; ML, medial lemniscus; MCP, medial cerebellar peduncle; IC, inferior colliculus; CIC, commissure of IC; LL, lateral lemniscus; PAG, periaqueductal grey; MGB, medial geniculate body; LGB, lateral geniculate body; SC, superior colliculus; RN, red nucleus; SN, substantia nigra; CST, corticospinal tract; St-DEC-TDI, short-tracks directionally encoded colour track density imaging; SNHL, sensorineural hearing loss.

We reconstructed the auditory pathway in vivo using probabilistic tractography in four subdivisions: the trapezoid body (TB), lateral lemniscus (LL), brachium of inferior colliculus (BIC), and acoustic radiation (AR) (see *Figure 2*). The auditory pathway was generally symmetric bilaterally. Except for the fibres connecting the CN to the contralateral SOC, contralateral probabilistic tracking resulted in fewer streamlines than ipsilateral counterparts (fibres tracking from the CN or SOC to the contralateral IC showed fewer than 30 streamlines each and were thus removed from further analyses).

## Tractography of the language pathway

The language pathway was also reconstructed bilaterally, each comprising two dorsal streams and two ventral streams (see *Figure 2*). Dorsal pathway I connects the posterior part of the superior temporal cortex (pSTC) to the premotor cortex (PMC) via the arcuate fascicle (AF) and the superior longitudinal fascicle (SLF). Dorsal pathway II connects the pSTC to the pars opercularis of Broca's area (BA44) via the AF/SLF. Ventral pathway I connects pars triangularis of Broca's area (BA45) and the temporal cortex via the extreme fibre capsule system (EFCS). Ventral pathway II connects the frontal operculum (FOP) and the anterior part of the STC via the uncinate fascicle (UF). The whole language pathway showed left dominance in fibre numbers.

## Children with profound SNHL exhibited fibre impairment in the central auditory pathway and the language pathway

In the central auditory pathway, FBA results demonstrated reduced fibre density (FD), fibre cross-section (FC), and fibre density and cross-section (FDC) in TB and decreased FC in LL in patients with profound SNHL (pFWE < 0.05) (see *Figure 3A*). There was no significant difference in the 'superior' part of the auditory pathway (i.e. the bilateral BIC and AR). In the language pathway, only the left ventral streams showed reduced FC and only the left dorsal streams showed reduced FD, while decreased FDC was found in both the left dorsal and left ventral streams (see *Figure 3B*). The size of impaired areas in the dorsal streams was larger than that in the ventral streams. No significant difference was found in fibre metrics of the right language pathway.

Next, we performed a tract-of-interest analysis to examine specific subdivisions in these pathways (see *Figure 3C and D*). Fibre metrics of all tracts displayed a decreased trend in children with

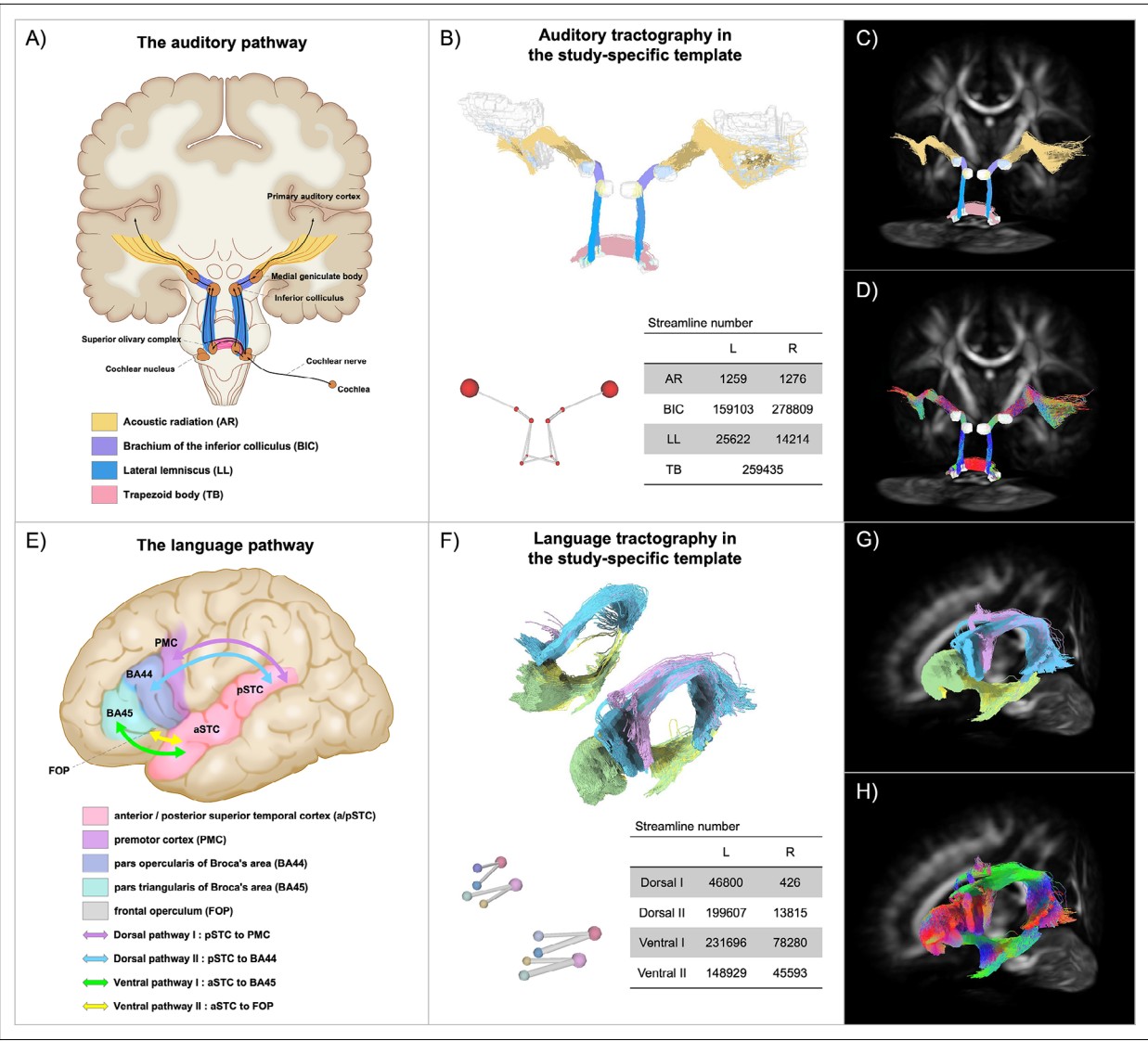

**Figure 2.** In vivo tractography of the auditory pathway and the language pathway in the study-specific template. (**A, E**) present schematic diagrams for the ascending auditory pathway and the language pathway, respectively. The auditory diagram was adapted from *Duvernoys Atlas of the Human Brain Stem and Cerebellum (2009)*; the language diagram was adapted from *Friederici et al., 2017*. (**B, F**) show three-dimensional reconstructions of tractography results of the central auditory pathway and the language pathway, respectively, in the study-specific template (at group level averaged from all 10 normal-hearing controls). Fibre colours refer to the subdivisions in each pathway, corresponding with the schematic colours in (**A**) and (**E**). The ball-and-stick diagrams represent the relative region-of-interest (ROI) size and streamline numbers in each pathway. Tractography results are also displayed in the study-specific white matter fibre orientation distribution (FOD) template in (**C**) and (**G**), colour coded by subdivision, and in (**D, H**), colour coded by direction (red: left-right; green: anterior-posterior; blue: superior-inferior). See *Figure 2—video 1* and *Figure 2—video 2* for three-dimensional animated videos of these pathways.

The online version of this article includes the following video(s) for figure 2:

**Figure 2—video 1.** Three-dimensional animated video of the central auditory pathway.
https://elifesciences.org/articles/85983/figures#fig2video1

**Figure 2—video 2.** Three-dimensional animated video of the language pathway.
https://elifesciences.org/articles/85983/figures#fig2video2

profound SNHL. In accordance with fixel-wise comparison results, only the 'inferior' subdivisions of the central auditory pathway (TB and bilateral LL) had significant fibre impairments. In the language pathway, significant impairment was found bilaterally rather than only on the left side. The left ventral I, left ventral II, and right dorsal II streams exhibited all-round fibre impairment, as FC, FD, and FDC were all significantly reduced. When considering central pathways as a whole, the mean FC of the

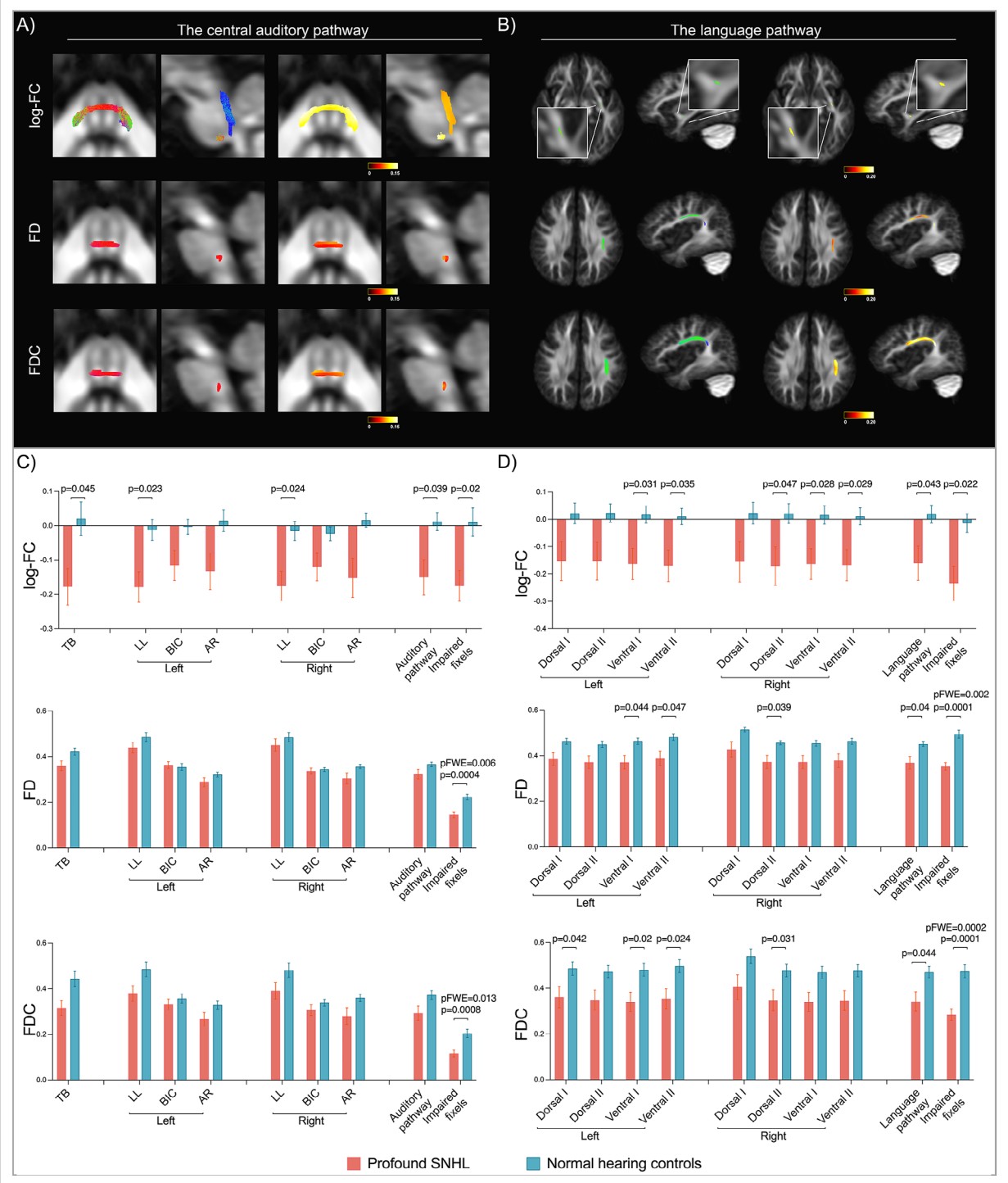

**Figure 3.** Fibre impairment of the central auditory pathway and the language pathway in children with profound sensorineural hearing loss (SNHL). Streamlines associated with significantly reduced fibre cross-section (FC), fibre density (FD), and fibre density and cross-section (FDC) (family-wise error [FWE]-corrected p-value<0.05) in fixel-wise comparison between patients with profound SNHL (n=13) and normal hearing controls (n=10) are shown for the central auditory pathway (**A**) and the language pathway (**B**). The left two columns in each panel display colour coded by direction and the right two coded by absolute values of effect size. Mean FC, FD, and FDC extracted from pathway subdivisions, entire pathways, and impaired fixels from (**A**) and (**B**) in the central auditory pathway (**C**) and the language pathway (**D**) are shown for patients with profound SNHL versus normal hearing controls. 'p' represents the uncorrected p-value; 'pFWE' denotes the FWE-corrected p-value. Non-significant p-values, whether uncorrected or FWE-corrected, are not displayed in the figure. The error bars represent Standard Error of the Mean. Refer to **Figure 3—figure supplement 1** for results displayed by

*Figure 3 continued on next page*

*Figure 3 continued*

separating three groups: profound SNHL with ear malformations and/or cochlear nerve deficiency (IEM&CND), profound SNHL with normal peripheral structure, and normal hearing controls.

The online version of this article includes the following figure supplement(s) for figure 3:

**Figure supplement 1.** Mean fibre cross-section (FC), fibre density (FD), and fibre density and cross-section (FDC) extracted from pathway subdivisions, entire pathways, and impaired fixels from *Figure 3* (**A**) and *Figure 3* (**B**) in the central auditory pathway and the language pathway of subjects from three groups: profound sensorineural hearing loss (SNHL) with ear malformations and/or cochlear nerve deficiency (IEM&CND) (n=6), profound SNHL with normal peripheral structure (n=7), and normal hearing controls (n=10).

entire central auditory pathway and the mean FC, FD, and FDC of the entire language pathway were significantly reduced. After extracting the mean values of significant results from fixel-wise comparison, the mean fibre metrics of these impaired fixels demonstrated a significant, large decrease. Of all tract-of-interest comparisons, only the decrease of FD and FDC of impaired fixels in both pathways survived family-wise error (FWE) correction ($p_{FWE} < 0.05$). It is worth noting, however, that in our tract-of-interest analysis, there is the potential for double dipping as we re-analysed the impaired pixels identified from the same comparison, which could have impacted the reported effect size.

## Peripheral nerve structure moderated the structural development of central pathways

In the present study, all seven children who underwent CI surgery had normal inner ear and cochlear nerve structure (see *Table 1* and *Figure 4A*). All six children who underwent ABI surgery or were ABI candidates presented with IEM&CND (see *Table 1* and *Figure 4B*). The cochlear nerve and vestibular nerve converge to form cranial nerve VIII (the vestibulocochlear nerve; cn.VIII), which travels through cerebrospinal fluid (CSF) in the cerebellopontine angle cistern and enters the brainstem. We measured the median contrast value of cranial nerve VIII (regressing out surrounding CSF median contrast values) to represent peripheral nerve tissue density (see *Figure 4C*), which provides a quantitative assessment of peripheral nerve structure. One patient with absent cochlear nerve also presented with no vestibular nerve; therefore, he was absent of cn.VIII and was left out in the following statistics.

We divided patients with profound SNHL into a normal peripheral structure subgroup and an IEM&CND subgroup, and compared their peripheral nerve and central pathway structure. The IEM&CND subgroup had significantly lower cn.VIII median contrast values than the normal peripheral structure subgroup ($p_{FWE} = 0.006$; see *Figure 4D*). Fixel-wise comparison showed no significant difference between the two subgroups in the central auditory pathway and the language pathway ($p_{FWE} > 0.05$). Then, we extracted mean fibre metrics of profound hearing loss-associated impaired fixels (significant regions in the fixel-wise comparison between patients with profound SNHL and normal hearing controls) and compared them between the two subgroups. All fibre metrics showed a reduced trend for the IEM&CND compared to the normal peripheral structure subgroup; only FD of profound hearing loss-associated impaired fixels in the language pathway showed a significant decrease but did not survive FWE correction (p=0.046, $p_{FWE} > 0.05$; see *Figure 4E*).

To investigate the relationship between peripheral nerves and central pathways, we performed a Pearson correlation between cn.VIII median contrast values and central pathway fibre metrics. When examining mean metrics of entire pathways, FD of the language pathway, rather than FD of the central auditory pathway, was significantly correlated with cn.VIII median contrast values (r = 0.57, p=0.032, $p_{FWE} = 0.194$; see *Figure 4F*). Similarly, for profound hearing loss-associated impaired fixels, the correlation between FD of the language pathway and cn.VIII median contrast values was stronger and more significant compared to the central auditory counterpart (r = 0.80 vs. r = 0.58, $p_{FWE} = 0.018$ vs. $p_{FWE} = 0.191$; see *Figure 4G*). No significant correlation was found in FC of entire pathways or profound hearing loss-associated impaired fixels. We also examined peripheral correlation with central pathway subdivisions. More correlations were found for language subdivisions than central auditory ones (see *Figure 4—source data 1*).

Further, a moderation analysis was conducted to explore the specific impact of peripheral nerve structure on the maturation of central pathways over time. The temporal developmental trajectories of FD of both the entire central auditory pathway and the entire language pathway were negatively moderated by cn.VIII median contrast values (interaction beta value = –0.379 and –0.298, respectively; see *Figure 4H and I*, *Figure 4—source data 2*). After controlling for gender, gestational weeks, and

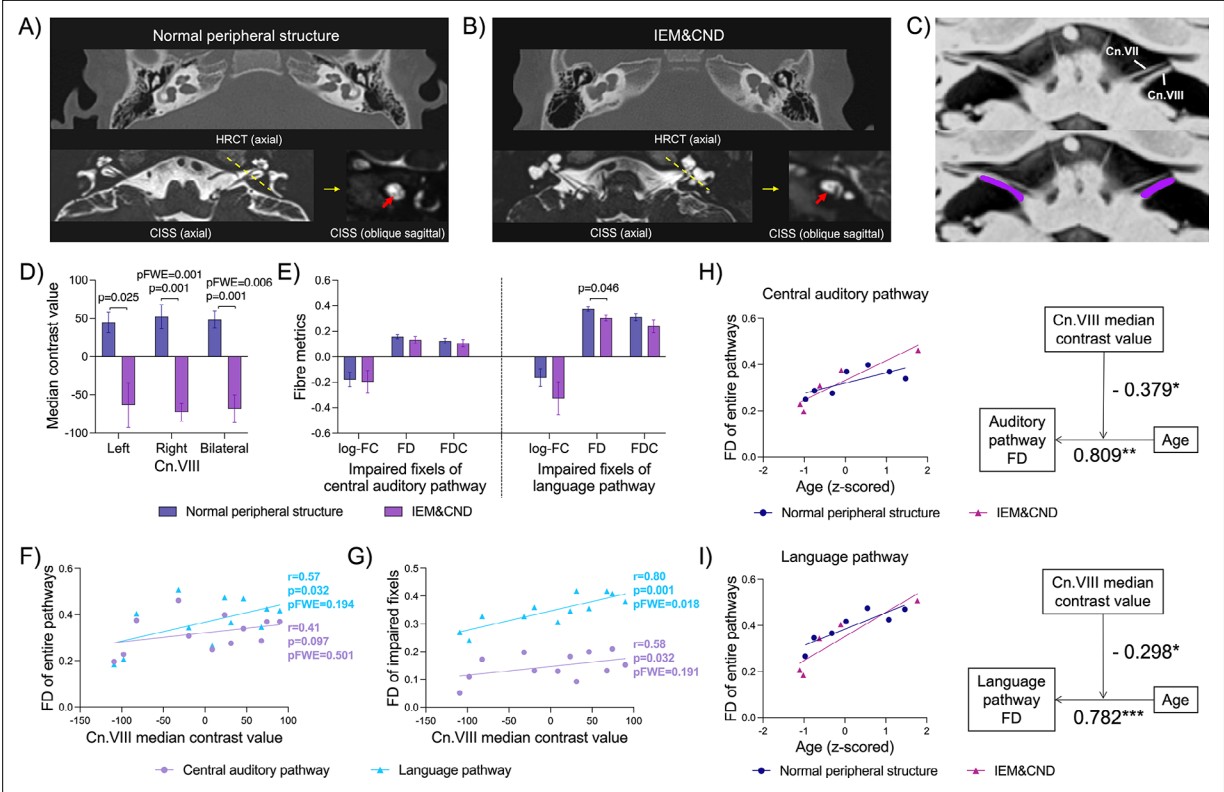

**Figure 4.** Cranial nerve VIII median contrast values and central pathway fibre metrics in profound sensorineural hearing loss (SNHL) patients with normal peripheral structure (n=7) or those with ear malformations and/or cochlear nerve deficiency (IEM&CND) (n=5). (**A**) shows temporal bone high-resolution CT (HRCT) and constructive interference in steady state (CISS) sections from a patient who underwent cochlear implantation (CI) surgery (male, 32 months old) and demonstrates a normal structure of the inner ear and cochlear nerve. (**B**) presents temporal bone HRCT and CISS sections from an auditory brainstem implantation (ABI) candidate (male, 17 months old) and reveals IEM&CND. The red arrow in (**B**) points to the missing cochlear nerve that is normally present, as shown by the red arrow in (**A**). (**C**) displays inverted CISS sections (axial plane) at the pontomedullary junction from a patient with profound SNHL. Cranial nerve VIII (cn.VIII) is visualized as a hyperintense structure relative to the surrounding cerebrospinal fluid (CSF). Cn.VIII was segmented (purple) and extracted for its median contrast value, regressing out surrounding CSF median values. (**D**) shows the cn.VIII median contrast values for patients with normal peripheral structure versus those with IEM&CND. (**E**) presents the mean fibre density (FD), fibre cross-section (FC), and fibre density and cross-section (FDC) of profound hearing loss-associated impaired fixels (from fixel-wise comparison results between patients and controls; see *Figure 2A and B*) in central pathways for patients with normal peripheral structure versus those with IEM&CND. (**F**) displays the Pearson correlation between cn.VIII median contrast values and the mean FD of entire central pathways for patients with profound SNHL. (**G**) illustrates the Pearson correlation between cn.VIII median contrast values and the mean FD of profound hearing loss-associated impaired fixels in central pathways for patients with profound SNHL. (**H**) demonstrates the moderation of central auditory pathway maturation by cn.VIII median contrast values. The mean FD of the entire central auditory pathway was significantly associated with age (beta value = 0.809), and their association was negatively moderated by cn.VIII median contrast values (interaction beta value = –0.379). (**I**) shows the moderation of language pathway maturation by cn.VIII median contrast values. The mean FD of the entire language pathway was significantly associated with age (beta value = 0.782), and their association was negatively moderated by cn.VIII median contrast values (interaction beta value = –0.298). These moderation effects are visualized as separate correlation plots of central pathway FD and age for patients with normal peripheral structure and those with IEM&CND. In panels (**D–G**), 'p' represents the uncorrected p-value; 'pFWE' denotes the family-wise error (FWE)-corrected p-value. Non-significant p-values, whether uncorrected or FWE-corrected, are not displayed in the figure. The error bars represent Standard Error of the Mean. In panels (**H**) and (**I**), *p<0.05, **p<0.01, ***p<0.001. Refer to *Figure 4—source data 2* for detailed statistics of the moderation analysis, as well as moderation analysis results after controlling gender, gestational weeks, and birth weights.

The online version of this article includes the following source data for figure 4:

**Source data 1.** Pearson correlation between cn.VIII median contrast values and fibre metrics of central pathways.

**Source data 2.** Detailed statistics of moderation analyses.

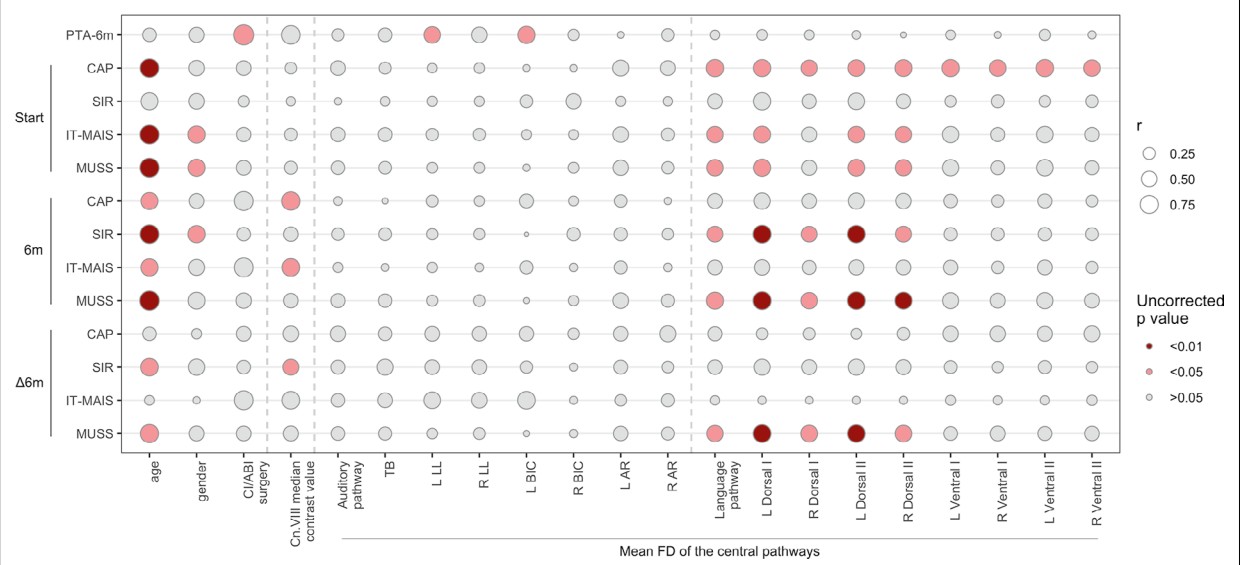

**Figure 5.** Correlations between preoperative characteristics and postoperative outcomes of auditory brainstem implantation (ABI) and cochlear implantation (CI) recipients. The size of the bubble represents the correlation levels; the colour intensity of the bubble represents the uncorrected significance levels. All correlations deemed significant are positive. Start, at device activation; 6 m, at 6 mo post-activation; Δ6m, changes over the 6-month intervening period; PTA-6m, pure tone average at 6 mo post-activation; CAP, Categories of Auditory Performance; SIR, Speech Intelligibility of Rating; IT-MAIS, Infant-toddler Meaningful Auditory Integration Scale; MUSS, Meaningful Use of Speech Scale. For correlations specific to CI recipients, please refer to *Figure 5—figure supplement 1*.

The online version of this article includes the following source data and figure supplement(s) for figure 5:

**Source data 1.** Postoperative outcomes of auditory brainstem implantation (ABI) and cochlear implantation (CI) recipients.

**Figure supplement 1.** Correlations between preoperative characteristics and postoperative outcomes of cochlear implantation (CI) recipients.

birth weight, the moderation effect was no longer significant (see *Figure 4—source data 2*). No significant moderation effect was found on the maturation of FC of central pathways by peripheral nerve structure.

## Investigating correlations between preoperative characteristics and postoperative outcomes

Our research aimed to explore the relationship between preoperative attributes and postoperative auditory and linguistic performance in all patients undergoing implantation (seven CI recipients and two ABI recipients). Evaluations included pure tone audiometry 6 mo post-device activation, and four auditory and language scales at both the onset and 6 mo post-activation, along with changes in these scores over the intervening period. The scales we employed were the Categories of Auditory Performance (CAP), Speech Intelligibility of Rating (SIR), Infant-toddler Meaningful Auditory Integration Scale (IT-MAIS), and Meaningful Use of Speech Scale (MUSS). These scales predominantly offer a qualitative and observational measure of auditory performance, speech use, and speech intelligibility in various real-life scenarios and are rooted in subjective interpretations. A notable trend was that subjective assessments of auditory and language development, as measured by various scales, appeared to be more responsive to preoperative metrics such as age, gender, choice of surgery, peripheral nerve tissue density, and fibre densities in the central auditory and language pathways. Conversely, the objective indicator of postoperative auditory condition (PTA) exhibited a more focused correlation pattern (see *Figure 5*).

Significantly, age showed a positive correlation with numerous auditory and language scales at both the initial and six-month post-activation, and with changes in these scores over this period. Gender demonstrated a correlation with IT-MAIS and MUSS at startup and with SIR 6 mo post-activation. This positive correlation implies that female patients (coded as '2') typically achieve higher scores on these scales than their male counterparts (coded as '1'). Contrasting with age and gender, the choice of surgery (CI versus ABI) manifested a unique positive correlation with

PTA 6 mo post-activation, but not with auditory and language scales. This suggests a more immediate effect on the objective auditory condition. A positive correlation signifies that the patients who opted for ABI surgeries (coded as '2') typically exhibited higher PTA values (indicating poorer hearing) compared to those who underwent CI surgeries (coded as '1'), at least at the 6-month post-activation.

In terms of the preoperative variables related to the auditory system, the cn.VIII median contrast value was positively correlated with CAP, IT-MAIS at 6 mo post-activation, and change in SIR. Interestingly, the FD of the central auditory pathway did not present significant correlations with the scales at any point.

However, a notable trend emerged when assessing the language pathways. CAP at the onset positively correlated with all language pathways, whereas PTA at 6 mo showed no correlation with these pathways. Generally, the FD of most language pathways, particularly most dorsal streams, demonstrated positive correlations with various scales at startup and 6 mo post-activation, and also with MUSS change. These observations highlight the language pathway's significant role in auditory development beyond the primary hearing level. When comparing the correlations between auditory and language pathways, it becomes apparent that the latter might exert a superior influence on auditory-speech outcomes. This could be attributed to the intricate and compensatory neural mechanisms involved in speech comprehension, especially in patients with auditory implants. Further exploration of these findings is elaborated upon in the 'Discussion section.

Despite the uniformly positive and frequently robust correlations ($r > 0.5$, many $> 0.75$), most did not pass the stringent FWE correction, and only managed to exceed uncorrected p-value thresholds. The only correlations that survived the rigorous FWE correction were those between age and IT-MAIS at startup, age and MUSS at 6 mo post-activation, and surgery choice and PTA at 6 mo. These findings warrant careful interpretation.

In addition, our findings presented an unexpected correlation: a positive association was identified between PTA at 6 mo post-activation and the FD of left LL and left BIC within the central auditory pathway. Considering that elevated PTA scores signify a degraded auditory condition, the implication of a positive correlation with the FD in these regions of the central auditory pathway raises a conundrum. Upon further reflection, we suspected that this positive correlation might be an artefact of statistical anomaly due to the polarized distribution of PTA values (see *Figure 5—source data 1*). Specifically, the PTA values of the eight patients analysed were situated at either high or low extremes, leaving few intermediate scores. The higher PTA values for ABI recipients, in particular, might skew the overall correlation. To substantiate this theory, we conducted a separate correlation analysis confined to CI recipients (see *Figure 5—figure supplement 1*). The results showed that the earlier identified positive correlation between PTA and other metrics vanished, thus corroborating our initial hypothesis. We therefore infer that the previously observed positive correlation between PTA and FD in certain regions of the auditory pathway could be a statistical illusion triggered by extreme PTA values in ABI recipients, rather than a genuine relationship. Conversely, the scale scores at both initial and 6-month timepoints did not appear as polarized as the PTA. On reviewing these results, we noted that the overall correlation pattern of the scales established in the all-patient cohort were largely preserved, such as the strong correlations with age and the association between language pathways and the majority of scales at both timepoints.

## Discussion

In this study, we successfully segmented subcortical auditory regions and reconstructed the auditory and language pathways in vivo. Our findings showed decreased FD and FC mainly in the inferior part of the central auditory pathway and the left language pathway. Additionally, we discovered that the correlation between language pathway fibre metrics and peripheral vestibulocochlear nerve structure is stronger and more significant than that in the central auditory pathway, and that the peripheral nerve structure moderated the developmental trajectory of the central auditory and language pathways. Preoperatively evaluating the structure of the auditory-language network helps predict postoperative audiometric and qualitative language outcomes.

## The new pipeline for mapping the human auditory pathway with in vivo MRI

To address the issue of inability to precisely locate subcortical auditory nuclei using sound-stimulating fMRI tasks for patients with profound hearing loss, we have introduced a new pipeline that only requires the acquisition and postprocessing of structural and diffusional images. The CN appears as an angulated wedge shape along the brainstem surface when viewed from above; it is mostly located between the inferior cerebellar peduncle (ICP) and CSF, with a small width of up to around 2 mm (*Rosahl and Rosahl, 2013*). The SOC is a small cell mass that is buried deep in the brainstem, surrounded by multiple fibre bundles. These two delicate auditory nuclei are prone to partial volume effects due to insufficient resolution and a lack of contrast to differentiate them from their surroundings in in vivo structural MR brain scans (even when using 7T MRI scanner; *Sitek et al., 2019*). To address this, we have reconstructed super-resolution TDI maps to provide complementary contrast for high-resolution structural images, allowing the CN and the SOC to be delineated according to track density and track direction at an isotropic resolution of 0.5 mm. The super-resolution properties of TDI maps have been validated using in vivo and in silico data, and the anatomical contrast of TDI maps from ex vivo mouse data has been related to histology (*Calamante et al., 2011*; *Calamante et al., 2012*). TDI maps have been demonstrated to be useful in delineating substructures of the thalamus, the basal ganglia, and brainstem fibre bundle cross-sections to assist fibre tracking (*Calamante et al., 2013*; *Kwon et al., 2021*; *Tang et al., 2018*). Epprecht et al. found that significantly different fibre orientations can be detected in the CN area and in the ICP area using diffusion tensor model (*Epprecht et al., 2020*), which suggests the potential role of fibre direction information in distinguishing the CN from adjacent structures. In the present study, the good inter-rater reliability of segmentation suggests that this method is effective for locating the subcortical auditory nuclei.

Our results showed that the best way to capture fibre features of different tracts in the auditory pathway is to track them separately and optimize tracking strategies for each part. We implemented probabilistic tractography and deliberately chose seeds and termination ROIs for each tract based on anatomical prior knowledge. MCP fibres were excluded when tracking the TB, which accounted for a large proportion of contralateral streamlines if not controlled and have yet been neglected in earlier studies. Exclusion ROIs are also essential when tracking the AR, because the AR crosses with several major fibre bundles that own greater FOD amplitudes in the corresponding directions. Mafei et al. found that optimal tracking parameters for the ARs were probabilistic tractography with default settings (angle threshold = 45°, step size = 1/2 * voxel size); however, the FOD amplitude threshold was not mentioned (*Maffei et al., 2019*). We examined a range of cutoff values and found that thresholding at 0.05 obtained optimal results for the ARs and 0.1 for subcortical auditory tracts. The tractography parameters established in the present study offer reliable recommendations for future attempts to track the auditory pathway.

## Fibre impairment pattern in the central auditory pathway of children with profound SNHL

Normally, the axonal myelination of the auditory pathway starts at the 26th foetal week, becomes definitive by the 29th week of gestation, and continuously increases in density until at least 1 y post-natal age (*Moore et al., 1995*). With the deprivation of hearing inputs, the central auditory pathways of children with congenital profound hearing loss displayed a brainstem-dominant fibre impairment pattern that included both microstructural impairment and macroscopic deficiency: the FD, FC, and FDC of the TB and the FC of the lateral lemnisci were significantly decreased; nevertheless, no significant difference was found in the branchium of inferior colliculus or the AR.

Current results are partly contradictory to earlier findings using the diffusion tensor model. Huang et al. manually delineated six ROIs along the auditory pathway, including nuclei and fibre bundles, using a 14 mm$^2$ square box in T2w and FA maps of children with profound hearing loss. They measured DTI-based metrics in the TB, SOC, IC, MGB, AR, and white matter of Heschl's gyrus (LL and BIC not included) and found decreased FA values in all of these six ROIs (*Huang et al., 2015*). Wu et al. focused on the AR and STG, demonstrating reduced FA values in both regions in children with hearing loss (*Wu et al., 2016*). The discrepancy between the present study and earlier findings may be caused by methodology factors. Unlike DTI metrics, FBA is capable of measuring individual fibre properties in fibre-crossing areas, which is the case along almost the entire auditory pathway. Based on our new

pipeline, each fibre bundle in the auditory pathway was defined by tractography separately; FBA metrics were measured in the fixel mask built on the auditory pathway so that presumable contamination from adjacent crossing fibres was excluded. Also, the FWE correction used in FBA is a more rigid multiple comparison method. As a result, we were able to make an accurate and strict examination of the fibre properties of the auditory pathway.

The TB was the most affected tract in the auditory pathway of children with congenital profound hearing loss. The TB is essential to sound localization because it transfers binaural sound information to the SOC, where interaural level and time differences are compared to identify the sound source. Decreased FD and FDC in the TB indicate an immature myelination or axon loss, which may affect the efficiency of signal transmission. For patients with bilateral cochlear implants, their accuracy and sensitivity of sound localization are still worse than those of normal hearing listeners and hearing aid users (*Dorman et al., 2016*; *Verschuur et al., 2005*). One reason for this may be the absent temporal fine structure cues and limited absolute level judgements in the CI system. Another possibility is that the structural impairment of the TB plays a role in the process. If this is true, it raises the interesting question of whether auditory implants may promote structural plasticity in the TB that could enhance sound localization performance over time.

The formation of a brainstem-dominant impairment pattern warrants investigation. The absence of a significant difference in two high-level auditory pathways, the BIC and AR, may reflect that the susceptibility of fibre structures to auditory deprivation decreases in a bottom-up fashion along the pathway, or that upper pathways in the thalamus and cerebrum may have already undergone cross-modal plasticity so that fibres are still structurally intact but subserve other functions. It has been proposed that the auditory cortex might reorganize to mediate other functions such as vision. Auditory areas in the superior temporal sulcus show greater recruitment in individuals with severe-to-profound hearing loss than in hearing individuals when processing visual, tactile, or signed stimuli (*Bavelier et al., 2006*). However, it is unclear that to what extent the primary auditory cortex may be affected by cross-modal plasticity and whether such cortical plasticity would affect downstream auditory nuclei and in-between fibre bundles. Further studies incorporating structural and functional features of the auditory pathway in larger samples are required to shed light on these questions.

## The structural development of the language pathway in children with profound SNHL

After CI and/or ABI implantation, children were able to achieve good hearing sensation, but their performance in speech recognition and production was poorer and varied (*Sennaroğlu et al., 2016*). One of the most important tasks in this field is identifying the factors that contribute to language outcome after implantation. In this study, we focused on the structural properties of the language pathway and found that patients with profound SNHL had an all-streams-affected fibre impairment that was left-dominant.

The language pathway was reconstructed based on Angela Friederici's model, which elucidates the neuroanatomical fibre bundles that underlie specific language functions (*Friederici, 2011*; *Friederici et al., 2017*). The coordination between BA44 and pSTC subserves syntactic computation, while the ventral streams that connect BA45 and FOP with the temporal cortex support lexical-semantic comprehension. The pSTC integrates syntactic and semantic information for comprehension at the sentence level and is also connected to the PMC as a peripheral sensorimotor interface system. The maturational status of these tracts has been found to be associated with behavioural performance. For example, the tract targeting BA44 is highly predictive of behavioural performance on processing hierarchically complex sentences (*Skeide et al., 2016*). In this study, these four streams were separately reconstructed using probabilistic tractography. FBA results showed that the dorsal streams mainly suffered from reduced FD, while the ventral streams mainly exhibited decreased FC in children with profound SNHL. The combined metric FDC was more sensitive in detecting fibre impairment in both dorsal and ventral streams. These results demonstrate that deprived auditory inputs have a damaging effect on the structural development of all language streams that serve different functions, which may indicate that children with congenital profound hearing loss suffer from an overall underdeveloped language capacity in semantics, syntax, and sensorimotor integration, rather than just struggling with oral communication.

## Moderation of central pathway maturation by peripheral auditory structure

Patients with normal peripheral auditory structure and those with IEM&CND had significantly different cn.VIII median contrast values (which has been shown to represent nerve tissue density; *Harris et al., 2021*), but few differences in FD and FC of the central auditory pathway and language pathway. However, several central pathways were positively correlated with cn.VIII median contrast values, including FC of bilateral BIC and LL, FC of two left ventral streams, and FD of bilateral dorsal streams and left ventral streams. Although all patients included in the study were diagnosed with bilaterally profound SNHL, several patients with normal peripheral structure had late-onset or progressive hearing loss that may have allowed a small amount of peripheral auditory inputs to stimulate central development before profound hearing loss. In contrast, all patients with CND failed newborn hearing screening and remained profound hearing loss since (see *Supplementary file 1*). This phenomenon may partly explain the correlation between peripheral nerve tissue density and central fibre metrics.

Our findings revealed a stronger and more significant correlation between language pathway fibre metrics and peripheral nerve tissue density than with central auditory metrics. This might seem counterintuitive given the structural and functional connections between the central auditory pathway and the peripheral cochlear nerve. For patients with profound SNHL, who present normal cochlear nerves and inner ear structures, the primary pathology is often located at the cellular level within the cochlea. However, in non-syndromic profound SNHL patients exhibiting IEM&CND, more severe genetic abnormalities are typically suspected. In our study cohort, genetic testing results were acquired for seven participants, revealing mutations in GJB2 (two patients), OTOF (one patient), and MYO15A (one patient). These mutations are largely associated with peripheral auditory deficits, given these genes' critical roles in inner ear development and function (*Kelsell et al., 1997*; *Stelma and Bhutta, 2014*; *Wang et al., 1998*; *Yasunaga et al., 1999*). Consequently, we hypothesized that observed central auditory pathway alterations could be adaptive changes due to these peripheral deficits. Nevertheless, it remains uncertain whether the detected central auditory anomalies stem from genetic factors directly or whether they are the result of underdevelopment due to auditory deprivation following peripheral dysfunction. Our findings suggest a weak association between profound hearing loss-related impaired fixels in the central auditory pathway and cn.VIII tissue density, casting doubt on the former hypothesis. It is important to consider, however, the potential of certain gene mutations such as CDH23 or CHD7 – typically linked with syndromic forms of hearing loss – to directly affect central auditory pathways (*Astuto et al., 2002*; *Zentner et al., 2010*). Although our study primarily included non-syndromic hearing loss patients, these potential genetic influences highlight the complexities inherent in interpreting auditory function and call for further in-depth investigation.

On the other hand, the significant association between FD/FC of profound hearing loss-associated impaired language fixels and cn.VIII tissue density suggests that the language pathway is more sensitive to peripheral auditory condition compared to the central auditory pathway. The language pathway, as the substrate for higher cognitive functions, develops in a way that is highly dependent on external inputs and interaction with the environment (*Kuhl and Rivera-Gaxiola, 2008*). More specifically, acquiring proper inputs during a sensitive period is essential for the maturation of the language pathway. This structural developmental feature corresponds with behavioural results from follow-up studies on CI recipients of different age at implantation. Chen et al. compared performances of children implanted before and after 2 years old and found that the younger the age of bilateral cochlear implant surgery, the higher developmental quotient of language, social skills, and adaptability the child could achieve after 2 y (*Chen et al., 2023*). Houston et al. studied an earlier demarcation timepoint at 1 year old and found that children implanted during the first year of life had better vocabulary outcomes than children implanted during the second year of life; however, earlier implanted children did not show better speech perception outcomes than later implanted children. They suggested that there is an earlier sensitive period for developing the ability to learn words than for central auditory development (*Houston and Miyamoto, 2010*).

To understand the specific impact of peripheral nerve structure on the developmental trajectory of central pathways, we conducted a moderation analysis. Cn.VIII tissue density had a significant negative moderation effect on the relationship between auditory pathway FD and age, and that between language pathway FD and age. In other words, the lower the cn.VIII tissue density, the closer the relationship between FD of central pathways and age. It is worth noting that cn.VIII tissue density

only moderated FD maturation; there was no significant moderation effect on FC maturation. These results may imply that the macroscopic developmental trajectory of central pathways is similar among patients with various peripheral conditions, while at the microscopic level, peripheral nerve deficiency may lead to a delayed and slowed axon generation or myelination in central pathways, which may reflect as an increased association of central FD with age in the participants ranging from 6 months to 6 years old. However, when controlling gender, gestational weeks, and birth weights, the moderation effect was not significant, necessitating a cautious interpretation of the findings. Taken together, to prevent aberrant structural development in the auditory-language network in the absence of hearing, auditory interventions should be implemented as early as possible. Special attention should be paid to patients with CND. Early interventions would provide children with timely auditory inputs during the critical period of language development and lead to better long-term outcomes.

Indeed, the quantitative differences observed between the effects on the central auditory and language structures may be influenced by the varying ability to accurately image each pathway with current neuroimaging technologies. The central auditory pathway, largely situated in the brainstem, faces severe distortion in diffusion images due to the inherent properties of the echo-planar imaging (EPI) sequence used. Despite distortion correction methods, it is challenging to entirely eliminate these effects. In contrast, the language pathways, primarily located in the cerebrum, have less distortion, allowing for more accurate imaging. Additionally, the auditory pathways in the brain (e.g. the auditory radiations) intersect with numerous cerebral fibre tracts, complicating accurate tractography. While the Constrained Spherical Deconvolution (CSD) modelling approach we employed is currently the most advanced method for addressing crossing fibres, it is not entirely exempt from these issues. The language pathways, on the other hand, intersect with fewer fibre tracts, facilitating more precise reconstruction. Lastly, the language pathways are more extensive and possess more fibres than the auditory pathways. From a statistical perspective, this enables more 'sampling' when extracting average metrics such as FD, resulting in more stable outcomes. Therefore, the differences in our ability to visualize and quantify these pathways might have influenced the comparative effects observed. As neuroimaging technologies and methodologies continue to advance, we anticipate that our understanding of the differential effects of hearing loss on the central auditory and language structures will continue to evolve and improve. Another limitation is that the normal hearing control group did not undergo scanning with the CISS sequence, precluding the establishment of a baseline for comparing the development of auditory

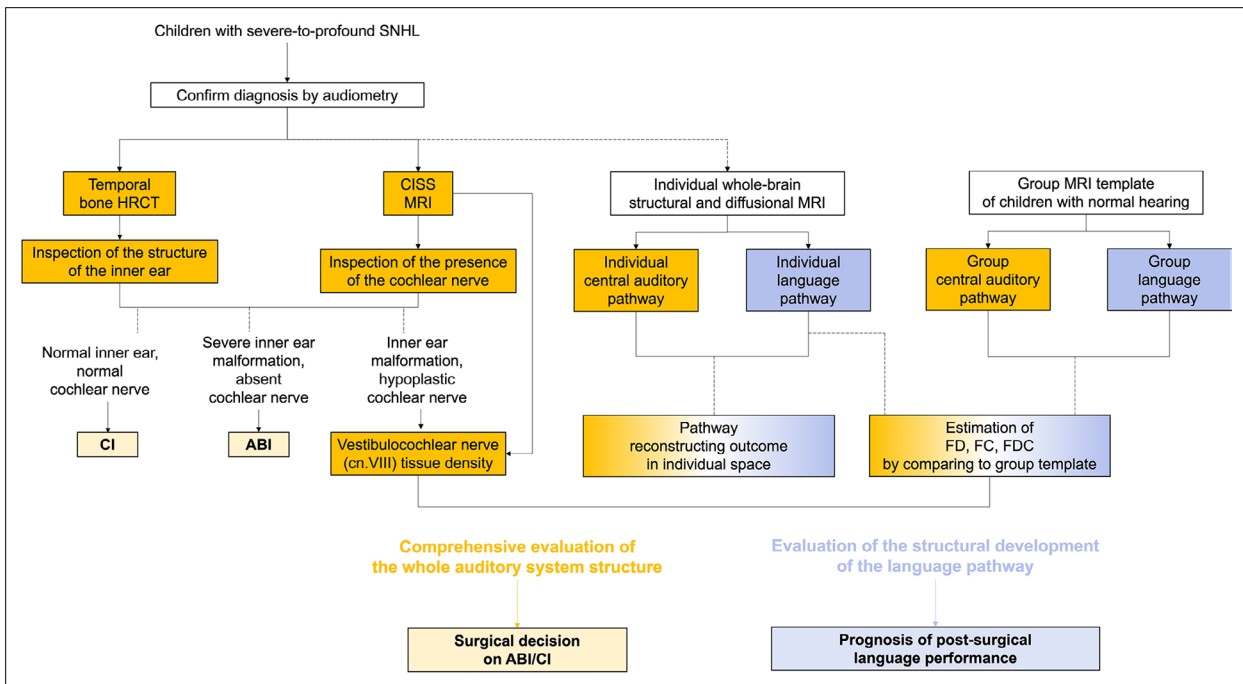

**Figure 6.** Comprehensive pre-surgical evaluation of the auditory-language network.

and language pathways in SNHL to normal development, which requires future research to include a baseline assessment.

## Integrating pre-surgical evaluations for optimized auditory-language rehabilitation

Our study delivers novel insight into the quantification of central auditory structural development and advocates an individualized, comprehensive pre-surgical evaluation of the auditory-language network to inform surgical decisions and predict prognosis (see *Figure 6*). From the correlation results, we observed significant clinical values that bolster the utility of such an evaluation. Further research should explore association and causation between neural and more robust auditory (e.g. speech-in-noise) and language evaluations.

### Comprehensive auditory system assessment: Informed surgical decisions

By refining the CISS-based estimation of cn.VIII tissue density, we enable more accurate quantitative assessment of peripheral nerve structure, which is essential for characterizing CND. The positive correlation of cn.VIII median contrast value with CAP and IT-MAIS at 6 mo post-activation, and with change in SIR, underscores its clinical value in predicting postoperative auditory performance.

Furthermore, by reconstructing the entire central auditory pathway, we enable measurements of FD and FCs to assess the structural maturity and integrity of the central auditory pathway. Although correlations with auditory-language scales were not significant in the all-patient analysis, among CI recipients, correlations were observed in specific regions of the auditory pathway. This suggests that this approach might be indicative of postoperative auditory information transmission efficiency and highlight the need for further investigations.

Moreover, by integrating the findings from HRCT, CISS MRI, and reconstruction of the central auditory pathway, we offer a holistic view from the cochlea to the auditory cortex, which may guide critical surgical decisions such as the choice between CI and ABI, and the side of implantation. Our study primarily includes patients with normal peripheral structures undergoing CI, but patients with varying malformations may also be CI candidates. Thus, while our findings provide initial insights, further investigations encompassing a more diverse patient population with different types of IEM&CND are warranted. It is also worth noting that the timepoint of outcomes post-implantation is important when comparing ABI and CI. Although our study found that ABI generally resulted in poorer PTA 6 mo post-activation compared to CI, clinical experience suggests that ABI patients may show considerable improvements over a longer term (within a year or two post-activation). Therefore, longer-term follow-up is essential to fully understand the outcomes of different surgeries. Our study's 6-month follow-up lays the groundwork for future, more comprehensive studies.

### Language pathways: Central to post-implant speech outcomes

We found that FD of most language pathways positively correlates with auditory and language scales. Our findings indicate the potentially central role of the structural development of the language pathway in predicting auditory and language outcomes following implantation.

Language has a recursive hierarchical structure, and the brain's mechanism of language processing reflects this through its hierarchical representations that unfold across time and space in the brain (*Ding et al., 2016*; *Friederici, 2011*; *Hickok and Poeppel, 2007*). Speech processing requires the transformation of continuous acoustic inputs into discrete linguistic constructs, termed the acoustic-phonological mapping. Many studies have found that Heschl's gyrus (HG) preferentially processes spectrotemporal features of sound, while STG preferentially processes phonological and intelligible features, indicating an ascending hierarchical progression from HG to STG (*Obleser et al., 2010*; *Peelle et al., 2010*; *Venezia et al., 2019*). However, recent research conducted by Edward Chang and colleagues demonstrated that transient functional disruptions or focal ablation of the HG did not impair speech comprehension, thereby suggesting a parallel processing between HG and STG (*Hamilton et al., 2021*). Furthermore, while STG is the computational hub for acoustic-phonological mapping, its response to speech stimuli is also modulated by cognitive factors such as attention, as well as high-level linguistic knowledge such as language familiarity and contextual information (*Leonard and Chang, 2014*; *Obleser and Kayser, 2019*). The dynamic interplay between STG and

higher-level centres such as the frontal cortex is crucial for speech processing (*Cope et al., 2017*; *Leonard et al., 2016*; *Sohoglu et al., 2012*). Hence, to some extent, STG in conjunction with these higher-level centres could be viewed as a speech processing network that operates independently of tonotopic mapping within HG. The language pathways reconstructed in this study form a significant part of this network.

Our findings revealed that subjective auditory-speech performance was predominantly correlated with the preoperative FD of the language pathway, rather than the structural status of the auditory system. This observation underscores the potential necessity of employing language-specific metrics to evaluate speech prognosis. It is noteworthy that sound information contains a significant redundancy for speech processing, as evidenced by the fact that people can understand degraded speech. In the context of patients with auditory implants, the received auditory signals tend to be unnatural and degraded. It is possible that the restoration of precise tonotopic mapping in these patients may not substantially influence their speech perception; instead, the dynamic interplay between STG and higher-order brain regions is instrumental in their speech comprehension capabilities. Predictions provided by higher-level centres aid patients in understanding and learning language. Peelle et al. found that listeners with cochlear implants showed greater activity in the left prefrontal cortex than listeners with normal hearing when listening to spoken words, indicating a compensatory role of the frontal lobe in language processing for implant recipients (*Sherafati et al., 2022*). As such, evaluating the structural integrity of connecting fibres between STG and higher-level centres preoperatively may predict the postoperative efficiency of information transmission within the network, which could be indicative of language-related behavioural outcomes.

Additionally, our results also indicate a significant positive correlation between age at the time of preoperative MRI scanning and postoperative performance. At first glance, this appears to be contradictory to the widely held view that there are advantages to performing surgery at an early age. However, it is important to note that the postoperative performance data in this study was gathered at 6 mo following surgery. Older children typically possess a more extensive repository of abstract concepts and might have a more developed top-down predictive function in higher-level brain areas. Consequently, in this short-term postoperative timeframe, older children may exhibit superior performance due to their advanced cognitive development. However, once the timeframe is extended, a wealth of behavioural evidence suggests that undergoing surgery earlier offers advantages in various areas, including auditory, linguistic, and cognitive development (*Chen et al., 2023*; *Friedmann and Rusou, 2015*; *The Joint Committee on Infant Hearing, 2019*). Therefore, in the context of clinical decision-making, it is advisable to perform surgery at the earliest feasible juncture. At this early stage, when the age range is narrow, evaluating the development of the language pathway preoperatively could hold substantial prognostic value.

Overall, while the study's findings emphasize the value of these preoperative metrics in determining postoperative outcomes, we acknowledge the need for additional research to corroborate these findings and unlock the full potential of these evaluations. By pushing forward in these investigations, we can continue to refine and enhance auditory rehabilitation, delivering patient-specific care and ultimately improving postoperative auditory and language outcomes.

## Strengths and limitations

The strengths of the present study include (1) the first comprehensive in vivo reconstruction of the central auditory pathway independent of using auditory stimuli; (2) the inclusion of children with IEM&CND in addition to those with normal inner ear structures to provide a more complete picture of children with congenital profound hearing loss; and (3) a precise inspection of fibre structures of the auditory pathway, the language pathway, and their subdivisions using a fixel-based approach that address crossing fibre issues.

We also acknowledge some limitations of this study: (1) the genetic dataset in our study is incomplete and heterogeneous. While two patients underwent extensive testing of over 200 genes, five were tested for select common genes. For a thorough understanding of the genetic factors in auditory deficits, future studies should employ uniform and comprehensive genetic testing across participants. (2) Our cohort is of a limited size, which may limit the generalizability of these findings to a larger population. (3) The potential discrepancies in the ability to image auditory and language pathways could have influenced the observed effects. Specifically, the more pronounced distortion

of the auditory pathway in diffusion images, the complex fibre-crossing issue within the auditory pathway, and the fewer fibres present compared to the language pathway may have contributed to the observed differences. (4) A 'floor effect' was observed in the PTA of ABI recipients and in some auditory and language scale scores. This may have masked the full extent of the relationships between the structural parameters and postoperative progress, as significant improvements could potentially be observed in some patients over a more extended period. (5) The scales employed (CAP, SIR, IT-MAIS, and MUSS) primarily provide subjective and observational measures. While they offer valuable qualitative insights into auditory performance and speech use in real-life contexts, they have limited quantitative precision. Their usefulness should be appreciated within this context.

## Conclusions

In conclusion, this study introduced a new pipeline for in vivo reconstruction of the central auditory pathway, found both microscopic and macroscopic fibre impairment in specific auditory and language tracts, and discovered a negative moderation effect of peripheral auditory structure on central pathway maturation. Additionally, our investigation uncovered significant correlations between fibre densities of auditory and language pathway subdivisions and various auditory and language outcomes, underscoring the potential predictive value of these structural parameters. This provides structural evidence supporting the necessity of early auditory intervention and establishes a promising comprehensive pre-surgical evaluation of the auditory-language network for children with severe-to-profound SNHL to assist with surgical planning and prognosis.

## Methods

### Study sample

Twenty-three children aged under 6 years old, including 13 patients with bilateral profound congenital SNHL and 10 controls matched on age and gender, were included.

The patients met the following criteria: (1) diagnosis of bilateral profound SNHL: click ABR threshold >95 dB nHL, or pure tone average (PTA, 0.5–4k Hz) threshold >95 dB HL (*Lin et al., 2011*); and (2) non-syndromic hearing loss (no association with any other systemic manifestations). The controls had normal hearing in both ears (click ABR threshold/PTA <20 dB HL). Exclusion criteria for all volunteers were as follows: (1) neurological disease (epilepsy, brain tumour, etc.) or history of head trauma; (2) psychiatric disorders (autism spectrum disorder, etc.); (3) history of ototoxic drug use; and (4) metal implants and other contraindications to MRI safety.

The study protocol was approved (SH9H-2021-T449-1) by the Ethics Committee of Shanghai Jiao Tong University School of Medicine Affiliated Ninth People's Hospital (Shanghai, China), and all enrolled subjects had informed consent provided by parent/guardian.

### Preoperative clinical and imaging evaluation

#### Clinical data

Age, gender, gestational weeks, birth weight, and a thorough medical history of pregnancy and hearing condition were recorded.

Pure tone audiometry was conducted at frequencies of 0.5k, 1k, 2k, 4k, and 8k Hz for children who were able to cooperate, serving to evaluate their hearing levels. Additionally, click ABRs were assessed. The click stimuli used in ABR tests had a duration of 100 μs and covered a broad frequency range from 100 Hz to 10,000 Hz. Examinations were performed using AC40 and Eclipse devices (Intercoustics, Middelfart, Denmark). Distortion product otoacoustic emissions (DPOAE) and tympanometry tests were also conducted to provide a comprehensive auditory assessment.

Genetic data were obtained from the patients' records, coming from various sources, which led to variations in the testing panels among patients. Specifically, two patients underwent comprehensive hereditary deafness gene testing, which analysed over 200 genes linked to hearing loss. In contrast, five patients were tested for a limited set of common genes, such as GJB2 and MTR. Genetic data for the remaining six patients could not be obtained due to practical constraints.

### CT, MRI acquisition, and quality assessment

A temporal bone high-resolution CT (sections of 0.5 mm in thickness) and a three-dimensional (3D) CISS MRI scan (3-Tesla MAGNETOM Vida, Siemens Healthcare, Erlangen, Germany) were acquired to

inspect the structure of the inner ear and the cochlear nerve of patients. Two experienced neuroradiologists assessed the presence and type of IEM&CND based on Sennaroğlu classification criteria (inter-rater agreement = 0.923) (*Sennaroğlu and Bajin, 2017*).

Whole-brain MRI was carried out on a 3-Tesla Siemens MAGNETOM Vida scanner (Siemens Healthcare) using a 64-channel head coil. To reduce motion artefacts, patients received sedation by oral intake of chloral hydrate for the MRI scan under supervision. Earplugs and sound-attenuating foam were used to decrease the noise of the scanner. Sandbags were placed on the scanner table to help attenuate vibrations during scanning. The lights in the scanner room were turned off, and blankets were wrapped around the children to create a suitable sleeping environment. Monitoring throughout the scanning session included pulse oximetry, respiration, and close observation by medical staff.

The acquisition protocol was adapted from the Developing Human Connectome Project (DHCP) that was optimized for the properties of the developing brain (*Edwards et al., 2022*). All children underwent a T1-weighted anatomical brain scan (3D MPRAGE sequence with a spatial resolution of 0.8 mm isotropic, matrix $= 320 \times 320$, field of view [FOV]$= 256 \times 256$ mm$^2$, and TR/TE $= 2400/2.38$ ms) and whole-brain diffusion imaging (dMRI). The scanning protocol consisted of 3 diffusion shells (b-values of 400, 1000, and 2600 s/mm$^2$) with 32, 44, and 64 diffusion-weighted directions each and 10 b0 volumes using PA phase encoding. Additionally, two b0 images in AP phase encoding were scanned for TOPUP distortion correction. The EPI readout used SMS factor of 4, Grappa acceleration factor of 2, partial Fourier factor of 6/8, isotropic resolution of 2.0 mm, axial slices of 72, matrix of $105 \times 105$, FOV of $210 \times 210$ mm$^2$, and TR/TE of 2900/95ms.

Each patient's MRI was transferred to a DICOM workstation during acquisition to review any clinical or research-relevant incidental findings. Then, insufficient coverage, excessive motion, and/or ghosting were visually assessed. If any image failed the visual quality assessment, the MR technicians would decide whether to re-scan the sequence according to the child's condition.

## Postoperative outcomes assessment
Of 13 patients, 9 underwent auditory implantation (two ABI recipients and seven CI recipients). Postoperative data were collected during follow-up appointments.

### Pure tone audiometry
This was conducted at 6 mo after device activation.

### Auditory and language scales
Four auditory and language scales were assessed both at activation and at 6 mo after activation. The scales included:

1. CAP: This scale categorizes the auditory performance on a spectrum from no awareness of environmental sounds to conversing with a stranger over the phone (*Archbold et al., 1995*). Mandarin version: test–retest reliability ($r = 0.981$, p<0.01), inter-rater reliability ($r = 0.983$, p<0.01), and criterion validity with LittlEARS ($r = 0.721$, p<0.01) and Griffiths ($r = 0.283$, p<0.05) (*Wang et al., 2020*).
2. SIR: The SIR assesses the intelligibility of speech, factoring in listener concentration, lip-reading cues, and context (*Allen et al., 1998*). Mandarin version: test–retest reliability ($r = 0.983$, p<0.01), inter-rater reliability ($r = 0.997$, p<0.01), and criterion validity with LittlEARS ($r = 0.698$, p<0.01) and Griffiths ($r = 0.428$, p<0.01) (*Wang et al., 2020*).
3. IT-MAIS: Centred on a child's capacity to respond to auditory stimuli without visual cues, the scale evaluates spontaneous use of sound in everyday settings and the ability to differentiate speech from non-speech sounds. Mandarin version: test–retest reliability ($r = 0.929$, p<0.01) and inter-rater reliability ($r = 0.894$, p<0.01). Content and construct validity were established with expert judgement and domain consistency (*Zhong et al., 2017*).
4. MUSS: MUSS evaluates a child's tendency to use speech as their primary communication mode in both familiar and unfamiliar settings, emphasizing understandability to both familiar individuals and strangers. Mandarin version: test–retest reliability ($r = 0.928$, p<0.01) and inter-rater reliability ($r = 0.910$, p<0.01). Its validity characteristics are in line with those of IT-MAIS (*Zhong et al., 2017*).

Assessments were conducted by certified audiologists and speech therapists, who were blinded to the preoperative evaluations to ensure unbiased data collection.

## MRI data analysis

### Cranial nerve VIII measurement

The measurement of cranial nerve VIII tissue density was adapted from Harris et al.'s protocol (*Harris et al., 2021*). Specifically, the contrast of CISS images was inverted to visualize cranial nerve VIII as a hyperintense structure relative to the surrounding CSF. The Cn.VIII was segmented on each axial section using ITK-SNAP (*Yushkevich et al., 2006*) by two independent raters. The inter-rater agreement was assessed using the DICE coefficient, yielding a value of $0.977 \pm 0.010$ (mean ± SD). The median contrast value of cn.VIII was calculated across sections. The median contrast values of adjacent CSF were also collected and regressed out to control contrast differences across individuals due to scanner heating and motion artefact.

### Diffusional MRI processing

The state-of-the-art FBA pipeline (*Dhollander et al., 2021*) was implemented to process diffusional data using MRtrix3 (*Tournier et al., 2019*). Specifically, after standard preprocessing steps (including denoising, Gibbs ringing correction, eddy-current and motion correction, bias field correction, and intensity normalization), response functions for WM, GM, and CSF were estimated from the data themselves. The diffusional images were then upsampled to 1.25 mm isotropic voxels for subsequent better estimation of fibre orientation distribution (FOD). We used multi-shell multi-tissue constrained spherical deconvolution (msmt-CSD) to obtain WM-like FOD as well as GM-like and CSF-like counterparts in all voxels (*Dhollander and Connelly, 2016*). A study-specific WM FOD template was created using the WM FOD images from all 10 controls. Finally, study-specific auditory and language pathways were generated from this template and filtered to reduce reconstruction bias (*Smith et al., 2015*). These generated tractograms were then converted to fixel masks, allowing for fixel-based analysis of specific tracts.

## Reconstruction of the human auditory pathway

### Overview

We generated directionally encoded colour track density imaging (DEC-TDI) maps from whole-brain tractography to obtain high spatial resolution images of the white matter. These DEC-TDI maps and T1-weighted images provided complementary information and enhanced anatomical contrast for subsequent manual segmentation of subcortical auditory nuclei. The primary auditory cortex was extracted from the Human Brainnetome Atlas (*Fan et al., 2016*) and co-registered to diffusional space. Finally, the auditory pathway was tracked based on anatomical prior knowledge and visualized using 3D volume rendering. This process was performed at both the group-average and individual level.

### Track density imaging

The DEC short-tracks TDI (stTDI) map method (*Calamante et al., 2012*) was used to obtain better directional information compared to the standard DEC-TDI pipeline (*Calamante et al., 2010*), particularly in low-intensity structures such as brainstem nuclei that we were interested in. Whole-brain probabilistic tractography was constrained to short tracks by setting the maximum length of each track to 20 mm (corresponding to 10 acquired voxels). We generated 40 million short tracks for each dataset using the iFOD2 algorithm (*Tournier et al., 2010*) by randomly seeding throughout the brain with the following parameters: angle threshold = 45°, minimum length = 4 mm, maximum length = 20 mm, cutoff value = 0.1. We then constructed the super-resolution TDI maps with a 0.5 mm isotropic grid size by calculating the number of tracks in each element of the grid. The colour-coding values were obtained by averaging the colours of all the streamline segments contained within each grid element, thereby indicating the local fibre orientation. (Green represents anterior-to-posterior, blue represents superior-to-inferior, and red represents left-to-right.)

## Image registration

At the group level, a study-specific T1-weighted brain template was created using the T1-weighted images from all 10 controls using antsMultivariateTemplateConstruction2.sh in ANTs (*Avants et al., 2011*) and transformed to the study-specific DEC-TDI space. At the individual level, T1-weighted images were transformed to each individual's DEC-TDI space. Therefore, manual segmentation can be carried out in T1-weighted and DEC-TDI with a 0.5 mm isotropic resolution.

## Manual segmentation of subcortical auditory regions

Subcortical auditory regions were segmented based on anatomical observations in histology studies (*Moore, 1987*; *Rosahl and Rosahl, 2013*; *Winer, 1984*) as well as earlier attempts to delineate some of these structures via MRI in vivo (*García-Gomar et al., 2019*; *Sitek et al., 2019*). Two raters independently segmented the auditory nuclei using the mrview toolbox in MRtrix3 (*Tournier et al., 2019*). Only the overlap areas between the two raters' segmentations were retained in the following analysis (inter-rater DICE coefficient: CN, 0.798; SOC, 0.690; IC, 0.829; MGB, 0.795).

### Cochlea nucleus (CN)

The CN is located on the brainstem surface at the pontomedullary junction, where auditory nerve axons enter and terminate. The CN is elongated and curved from ventrolateral to dorsomedial. The ventral and dorsal portions of the CN can be distinguished by histological cytoarchitectonic properties, although approximately 10% of their shared borders remain a grey zone (*Rosahl and Rosahl, 2013*). In a horizontal view, the CN borders the ICP medially; its anterior half extends laterally along the posterior edge of the middle cerebellar peduncle (MCP) (*Moore and Osen, 1979*; *Terr and Edgerton, 1985*).

On T1-weighted images, the CN can be located at the pontomedullary junction where cranial nerve VIII enters the brainstem and is roughly delineated along the brainstem surface from ventrolateral to dorsomedial. In DEC-TDI maps, the CN is distinguished from its medial neighbour, the ICP, by its clear blue border, as the ICP travels mainly in the rostrocaudal direction (*Epprecht et al., 2020*). The CN, on the other hand, is a mixture of cell bodies and axons that travel from ventrolateral to dorsomedial, resulting in either low-intensity areas (where cells dominate) or green colour areas (where axons dominate).

### Superior olivary complex (SOC)

The SOC is a group of cells located in the pons, a short distance medial and rostral to the CN. The SOC is composed of a laminar medial nucleus (which extends about 4 mm rostrocaudally) and a small lateral nucleus; the entire complex is enclosed by a capsule of rostrally directed axons of the ascending auditory pathway (*Moore, 1987*; *Strominger and Hurwitz, 1976*).

The SOC is not distinguishable on T1-weighted images. In DEC-TDI maps, the SOC appears as a hypointense area surrounded by hyperintense fibres in the horizontal view: medially, the medial lemnisci; ventrally, the TB; and laterally, the MCP. The SOC was delineated from the same axial plane as the rostral-most extent of the ventral CN, extending about 4 mm rostrally (*García-Gomar et al., 2019*).

### Inferior colliculus (IC)

The IC is easy to locate as the two inferior spherical structures of the corpora quadrigemina in the dorsal midbrain (*Mansour et al., 2019*). On T1-weighted images, the IC is distinguished from its medially adjacent structure, the periaqueductal grey matter, by demonstrating more intense T1 signals in the horizontal view. However, in DEC-TDI maps, most of the signal of the IC is lost, possibly due to distortion from tissue/air interface.

### Medial geniculate body (MGB)

The MGB is located in the ventromedial thalamus. On T1-weighted images, the MGB is identified as an oval-shaped hypointense eminence that is medial to the lateral geniculate body and lateral to the superior colliculus (*Winer, 1984*). In the horizontal view of DEC-TDI maps, the MGB is restricted ventrolaterally by the blue areas of the corticospinal tract (CST).

## Tractography

Subcortical auditory regions were segmented manually as described above. The primary auditory cortex (A41/42, TE1.0 and TE1.2) was extracted from the Human Brainnetome Atlas (*Fan et al., 2016*) in the MNI space and co-registered to the WM FOD space by a rigid, affine, and nonlinear transformation using ANTs (*Avants et al., 2011*). Probabilistic tractography of each major tract in the auditory pathway was performed in the WM FOD space using iFOD2 algorithm with optimized parameters. For cortical tracts (ARs), we set a 0.05 cutoff value and 80 mm maximum length; for subcortical tracts, the cutoff value was 0.1 and the maximum length was set to 200 mm. Other parameters were kept the same across all tracts: 10,000 seeds per voxel, angle threshold = 45°, and minimum length = 4 mm. A mask of brainstem and thalamus was created semi-automatically using ITK-SNAP (*Yushkevich et al., 2006*) to constrain tracking of the subcortical auditory pathway; a centre ROI in the sagittal plane was used as exclusion for ipsilateral tracking.

### Trapezoid body (TB) and lateral lemnisci (LL)

The neurons in the CN receive nerve innervation from the cochlea and project to the IC both directly and indirectly. The SOC is the major relay station and receives axons from mostly the contralateral, partly the ipsilateral CN; the contralateral dominance remains in the following ascending pathway (*Moore, 1987*). The ascending axons below the level of the SOC form the TB; axons above the level of the SOC form the LL (*Moore et al., 1995*). The TB travels horizontally in the inferior pons and comprises many crossing fibres. The LL runs rostrally and dorsally to the IC and is located on the lateral side of the brainstem superficially.

Tracking of the TB and LL was carried out by seeding from each CN to the SOC and the IC in both sides, using MCP as an exclusion ROI for anatomical constraints, and seeding from each SOC to the IC in both sides.

### Brachium of inferior colliculus (BIC)

All ascending projections from the auditory brainstem to the thalamus are carried in the BIC (*Moore, 1987*). Tracking of the BIC was performed by seeding from the IC to the ipsilateral MGB. There are also commissural pathways between the bilateral IC; however, the commissure of IC is difficult to trace via tractography due to signal loss near the tissue/air interface.

### Acoustic radiation (AR)

The AR is the final stream that links the subcortical auditory pathway to the auditory cortex (*Rademacher et al., 2002*). The AR crosses with or travels near many major fibre bundles on its way to the auditory cortex, including the CST, the arcuate fasciculus (AF), the inferior fronto-occipital fasciculus (IFOF), the middle longitudinal fasciculus (MLF), the inferior fasciculus (ILF), and the optic radiation (OR). We manually delineated major cross-sections of the tracts mentioned above as exclusion ROIs. Then, we tracked the AR by seeding from the MGB, terminating in the ipsilateral primary auditory cortex, and excluding the adjacent tracts for anatomical constraint.

## Visualization

Streamlines were transformed into the trk format in Python via the Nibabel package (https://nipy.org/nibabel/) and visualized in DSI Studio (https://dsi-studio.labsolver.org/).

## Reconstruction of the language pathway

Language ROIs were also extracted from the Human Brainnetome Atlas (*Fan et al., 2016*) for its finer subdivision in the temporal and frontal cortex, and co-registered from the MNI152 T1 space (*Fonov et al., 2009*) to the WM FOD space using rigid, affine, and nonlinear transformation with ANTs (*Avants et al., 2011*). The anterior superior temporal cortex (aSTC) was defined as the combination of A22r, A38l, and aSTS; the posterior superior temporal cortex (pSTC) was segmented by combining A22c, rpSTS, and cpSTS. The frontal areas were also extracted: the pars opercularis of Broca's area (BA44), pars triangularis of Broca's area (BA45), the FOP, and the PMC.

Probabilistic tractography was performed in the WM FOD space. Two dorsal streams of the language pathway were seeded from BA44 or PMC and terminated in pSTC; two ventral streams were seeded from BA45 or FOP and terminated in aSTC. Parameters included iFOD2 algorithm, 10,000

seeds per voxel, angle threshold = 45°, cutoff value = 0.1, minimum length = 4 mm, maximum length = 200 mm. Tracts were visualized in the same way as the auditory pathway.

### Fixel-based metrics

In the FBA framework, a 'fixel' refers to a 'fibre population within a voxel', allowing for the measurement of WM metrics for individual fibres crossing in the same voxel. FD, FC, and FDC were calculated for the study-specific auditory pathway and the study-specific language pathway. FD values are approximately proportional to total intra-axonal volume and measure WM microstructure, while FC estimates macroscopic differences by using information from individual subject warps to the study-specific template. The combined FDC measure enables a more sensitive assessment of fixel-wise effects (*Dhollander et al., 2021*).

### Statistical analysis

Fixel-wise comparison of FC, FD, and FDC between groups was conducted using the connectivity-based fixel enhancement method (*Raffelt et al., 2015*). Tract-wise analysis was performed by extracting the mean fibre metrics of tracts-of-interest and comparing them between groups using PALM (*Winkler et al., 2014*) in MATLAB. For both comparisons, age and gender were controlled, and family-wise-corrected (FWE) p-values were obtained via permutation testing. Pearson correlation between cn.VIII median contrast values and central pathway fibre metrics were conducted using PALM in MATLAB. Moderation analysis was performed using SPSS. Pearson correlation between preoperative characteristics (age, gender, surgical choice, cn.VIII median contrast values, mean FD of the central auditory pathway and the language pathway) and postoperative outcomes (PTA and four auditory and language scales) was also conducted using PALM in MATLAB.

## Acknowledgements

This research was funded by Science and Technology Commission of Shanghai Municipality Major Basic Research (2018SHZDZX05), Science and Technology Commission of Shanghai Municipality Shanghai Key Laboratory of Translational Medicine on Ear and Nose diseases (14DZ2260300), Shanghai Municipal Health Commission Shanghai Key Clinical Specialty Construction - Otolaryngology-Head and Neck Surgery (shslczdzk00802), Shanghai Shen Kang Hospital Development Center Emerging Frontier Project (SHDC12020105), Shanghai Jiao Tong University School of Medicine Translational Medicine Collaborative Innovation Project (TM202011), and Shanghai Jiao Tong University School of Medicine High-level Local University Construction Project. The funding sources were not involved in the design of the study, the collection, analysis and interpretation of data, writing the manuscript, or the decision to submit the manuscript for publication.

## Additional information

### Competing interests

Yinghua Chu: Yinghua Chu is affiliated with Siemens Healthineers Ltd. The author has no financial interests to declare. Yang Song: Yang Song is affiliated with Siemens Healthineers Ltd. The author has no financial interests to declare. The other authors declare that no competing interests exist.

### Funding

| Funder | Grant reference number | Author |
| --- | --- | --- |
| Science and Technology Commission of Shanghai Municipality | Major Basic Research 2018SHZDZX05 | Hao Wu |
| Science and Technology Commission of Shanghai Municipality | Shanghai Key Laboratory of Translational Medicine on Ear and Nose diseases 14DZ2260300 | Hao Wu |

| Funder | Grant reference number | Author |
| --- | --- | --- |
| Shanghai Municipal Health Commission | Shanghai Key Clinical Specialty Construction - Otolaryngology-Head and Neck Surgery shslczdzk00802 | Hao Wu |
| Shanghai Shen Kang Hospital Development Center | Emerging Frontier Project SHDC12020105 | Zhaoyan Wang |
| Shanghai Jiao Tong University School of Medicine | Translational Medicine Collaborative Innovation Project TM202011 | Hao Wu |
| Shanghai Jiao Tong University School of Medicine | High-level Local University Construction Project | Hao Wu |

The funders had no role in study design, data collection and interpretation, or the decision to submit the work for publication.

### Author contributions

Yaoxuan Wang, Conceptualization, Data curation, Formal analysis, Validation, Investigation, Visualization, Methodology, Writing - original draft, Writing – review and editing; Mengda Jiang, Conceptualization, Resources, Data curation, Formal analysis, Validation, Investigation, Visualization, Methodology, Writing – review and editing; Yuting Zhu, Formal analysis, Validation, Visualization, Methodology, Writing – review and editing; Lu Xue, Hongsai Chen, Validation, Writing – review and editing; Wenying Shu, Xiang Li, Visualization, Writing – review and editing; Yun Li, Ying Chen, Yu Zhang, Data curation, Project administration, Writing – review and editing; Yongchuan Chai, Xiaofeng Tao, Resources, Data curation, Supervision, Project administration, Writing – review and editing; Yinghua Chu, Conceptualization, Resources, Supervision, Funding acquisition, Methodology, Writing – review and editing; Yang Song, Data curation, Methodology, Project administration, Writing – review and editing; Zhaoyan Wang, Conceptualization, Resources, Data curation, Supervision, Funding acquisition, Validation, Investigation, Project administration, Writing – review and editing; Hao Wu, Conceptualization, Resources, Data curation, Supervision, Funding acquisition, Project administration, Writing – review and editing

### Author ORCIDs

Yaoxuan Wang http://orcid.org/0000-0003-1002-7242
Zhaoyan Wang http://orcid.org/0000-0002-0977-0920
Hao Wu http://orcid.org/0000-0002-5317-902X

### Ethics

The experimental procedures were approved by the Ethics Committee of Shanghai Jiao Tong University School of Medicine Affiliated Ninth People's Hospital (Shanghai, China) (reference number: SH9H-2021-T449-1), and were performed in accordance with the approved guidelines. All enrolled subjects had informed consent provided by parent/guardian.

### Decision letter and Author response

Decision letter https://doi.org/10.7554/eLife.85983.sa1
Author response https://doi.org/10.7554/eLife.85983.sa2

## Additional files

### Supplementary files

• Supplementary file 1. Audiological characteristics of patients with congenital bilateral profound sensorineural hearing loss.

• Supplementary file 2. Genetic profile of patients with congenital bilateral profound sensorineural hearing loss.

• MDAR checklist

## Data availability

The subcortical auditory segmentations of the study-specific template (as well as T1-weighted, DEC-TDI, and WM FOD templates) are available on the Open Science Framework: https://osf.io/pxmf5/.

The following dataset was generated:

| Author(s) | Year | Dataset title | Dataset URL | Database and Identifier |
|---|---|---|---|---|
| Wang Y, Jiang M, Zhu Y | 2023 | Mapping the auditory pathway | https://doi.org/10.17605/OSF.IO/PXMF5 | Open Science Framework, 10.17605/OSF.IO/PXMF5 |

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
