## [Editor Report]

This important study used high-resolution brain imaging methods to visualize and index non-invasively auditory and language pathways of young children born with inner ear malformations or cochlear nerve dysfunction resulting in profound hearing loss. Nerve fibre impairments were compellingly demonstrated in subcortical auditory and cortical language pathways relative to typical hearing controls. Qualitative language assessment and audiometry linked these structural findings with functional outcomes. The results suggested novel approaches for clinical assessment of central auditory and language pathways that may influence different intervention strategies.

---

## [Decision Letter]

**Decision letter after peer review:**

Thank you for submitting your article "Impact of Peripheral Auditory Structure on the Development of Auditory-Language Network in Children with Profound Sensorineural Hearing Loss" for consideration by *eLife*. Your article has been reviewed by 3 peer reviewers, and the evaluation has been overseen by a Reviewing Editor and Barbara Shinn-Cunningham as the Senior Editor.

Essential revisions:

1) Behavioral data on the speech and language abilities of the children are available and should be reported.

2) The clinical evaluation and presentation of the children, both genetically and auditorily, is inadequate. Please provide any further details that would be useful for interpreting the other results.

3) The distinction between the groups isn't adequately considered. Please provide further details.

4) The practical relevance of the auditory-language pathways is unclear. Aside from behavioral data (point 1, above), a broader consideration of the studied pathways in the overall processing of hearing and language by the brain is needed.

5) Add discussion of the possibility that the quantitative difference between effects on central auditory and language structures is due to the different ability to image each pathway.

*Reviewer #1 (Recommendations for the authors):*

Overall, the authors have done a beautiful job creating high-resolution maps of the auditory pathway in these children. This type of imaging is not my speciality and so I cannot comment on the methodology or analysis behind the imaging. However, I am an expert on pediatric hearing loss and so I evaluated this manuscript from that perspective.

Detailed recommendations are below:

1) There are some major deficiencies with respect to the clinical evaluation/presentation of the children in this study which makes interpretation of the results difficult. First of all, there is no diagnosis provided. The causes of pediatric hearing loss is extraordinarily complex with hundreds of genetic forms and environmental causes as well. Hearing loss is not a diagnosis but is instead a symptom of an underlying pathologic change to the peripheral and/or central auditory system. The authors do not provide a diagnosis for the 13 children with hearing loss and as such any conclusions are limited. For example, if some of the children have GJB2 hearing loss, I would expect differences compared to children with CDH23 hearing loss (which has been shown to have some central auditory effect). Similarly, variants in CHD7, are associated with CHARGE syndrome and are a common cause of cochlear nerve deficiency, with some presumed central effect. The authors should provide more clinical history on individual patients, including hearing testing, and should at the least have performed genetic testing for a full evaluation of hearing loss. Ideally, the authors would then separate their subjects into groups (i.e. those with GJB2 hearing loss, versus others). This would make their study more powerful.

2) This study does not include cochlear implant and/or auditory brainstem implant outcomes. Therefore, the authors cannot have any claims to determining or predicting outcomes with these devices until their correlate this finding with those results. This would be an excellent follow-up study and the authors should temper their statements on clinical applicability until this is completed.

3) I assume that the CIs and ABIs were implanted after imaging? This is not specified anywhere that I could find.

4) The figure legends are confusing. Please indicate whether each image is an individual patient or an average and which group each image belongs to.

5) The graphs in Figure 3 are very very difficult to read. The authors should reformat these so that there is a white background and the bars are filled (or at least not use the current color scheme).

6) The authors should be careful about how they use the word 'deaf'. 'Deaf or hard of hearing' is an appropriate way to describe individuals with hearing loss. Severe-to-profound or profound are accurate ways to describe the severity of a hearing loss. The use of the word 'deaf' to describe profound hearing loss is generally discouraged as it is not scientific and may be confused by 'Deaf' (capital D) which describes members of the Deaf community.

Please see above.

*Reviewer #2 (Recommendations for the authors):*

General comments

– Although the paper tried to justify why focusing on IEM and CND is important, such decision seems still arbitrary to this reviewer. More discussion on what makes these hearing loss etiologies distinct from other etiologies would help.

– The paper does not discuss how these results help with a better understanding of speech sound encoding and decoding in listeners with normal and impaired hearing. It is particularly important to discuss these findings in the context of mapping sound patterns to phonological units.

– Also, there is a rich literature on hearing, language comprehension, and speech perception network from Gregory Hickok, David Poeppel, and Jonathan Peelle which are not discussed in this paper.

Title

– The title needs to be more specific to the questions of this study. Using the general term "Peripheral Auditory Structure" could be misleading. Please revise to reflect both the specific types of hearing loss in this study and also the population that is children with profound hearing loss, but receive CI or ABI as interventions.

Abstract:

– Cochlear implantation and auditory brainstem implantation can provide hearing sensation… it is more accurate to say … provide "partial" hearing sensation …

– "Previous attempts to locate subcortical auditory nuclei using fMRI responses to sounds are not applicable to deaf patients." Please add a reason(s) for why these previous attempts are not applicable.

– Line 3: lifelong "negative" consequences …

– Line 39: please check the in-text citations format

– Line 52: "Secondly, although earlier studies reported several altered fibre tracts related to language function, …" Please add references to support this statement.

– Line 83: mean [SD], age, 30.92 [6.115] months; 9 males, 4 females

– Mean [SD] of age

– You can remove 4 females as it is redundant

– Please consider the same edit for NH

– Line 85: Please indicate the confidence interval in your stats report

Table 1

– I recommend using the term gender instead of sex

Line 137

– FC, FD, and FDC are presented here for the first time in the manuscript. So, their extended version should be used here to avoid possible confusion.

Caption of Figure 3: please report the actual p-values

– Line 139: ie. ◊ i.e.

Figure 3

– I wonder how the data and results would look if ABI and CI groups were represented separately.

– Line 167-168: please elaborate on why such surgical decisions have limited value.

– Line 193: Report the actual p–values

– Line 451: Provide more details on the nature of click stimuli including frequency.

– Line 453: Clarify the dB HL used for NH decision (20 dB HL?).

– Line 468: If these measurements were not used in this study, it is not clear to this reviewer why they are reported. Please clarify.

– Line 479–481: please add inter–rater reliability to provide evidence on the acceptable agreement between raters.

– Line 503–504: please report how many images were excluded in this process if any.

– Line 511: Does {plus minus}0.010 refer to standard deviation? If so, please indicate that in the paper.

– Table 1 and 2 in appendices: round correlation coefficients to two decimal places. More than two decimal places seem redundant.

*Reviewer #3 (Recommendations for the authors):*

Recommendations:

1. Enhance the reliability and replicability of research findings by increasing the sample size for both SNHL and NH children. The current sample size is insufficient for between-group comparisons and correlational analyses, particularly when examining sub-groups of SNHL children with only 6 or 7 subjects per group.

2. Incorporate post-surgical behavioral outcome measures into the analysis to assess the value of pre-surgical neuroimaging and neural analyses. This will help elucidate the relationship between pre-surgical neuroimaging and post-surgical behavioral benefits, informing clinical decision-making.

3. Reevaluate data analyses for potential double dipping issues, specifically in lines 153-155 and Figure 3 C&D. The impaired pixels identified by comparing SNHL and NH groups are re-analyzed for the same comparison, potentially inflating effect size by only examining the ROI/pixels derived from the same comparison.

4. To better understand age effects, conduct the analyses presented in Figure 4 for NH children, which will serve as a baseline to demonstrate how different types of SNHL modulate the development of auditory and language pathways in comparison to normal development.

5. Ensure that modulation effects displayed in Figure 4 are controlled for confounding variables, such as residual hearing, sex, gestational weeks, birth weight, etc. to improve the validity of the results.

---

## [Author Response]

Essential revisions:1) Behavioral data on the speech and language abilities of the children are available and should be reported.

Thank you for the insightful comment. We agree that the inclusion of post-implantation behavioral data on speech and language abilities is crucial to further support the value of our findings. We have now incorporated this data into our analysis, which can be found in the ‘Results’ section (see Page 15-17) and ‘Discussion’ section (see Page 23-25). Our analysis shows a promising correlation between the pre-operative MRI metrics and post-implant speech and language outcomes, suggesting the potential of these imaging markers in predicting outcomes.

2) The clinical evaluation and presentation of the children, both genetically and auditorily, is inadequate. Please provide any further details that would be useful for interpreting the other results.

Thank you for your comment. We agree that a more comprehensive clinical and genetic characterization of the children could add valuable context to our study. Accordingly, we have included more comprehensive clinical data, available genetic testing results, and auditory evaluation results for all children in the revised manuscript. Please see Supplementary file 1 and Supplementary file 2 for these updates. For more detailed information, especially about the genetic characterization and limitations of our study, please refer to our response to Reviewer 1's Comment 1 below.

3) The distinction between the groups isn't adequately considered. Please provide further details.

Thank you for your comment. We acknowledge the significance of clearly defining and justifying our choice of groups, particularly emphasizing Inner Ear Malformation (IEM) and Cochlear Nerve Deficiency (CND) to enhance the contextual understanding of our research. We have carefully reviewed reviewer 2's comment 1 and incorporated it into our revision process. We have added a comprehensive discussion on the significance of focusing on IEM&CND, the challenges it poses in clinical management, and the complexities associated with its etiology in the ‘Introduction’ section (Page 2, Line 15-43). These additions aim to provide a clearer understanding of the differentiation between the groups under study.

4) The practical relevance of the auditory-language pathways is unclear. Aside from behavioral data (point 1, above), a broader consideration of the studied pathways in the overall processing of hearing and language by the brain is needed.

Thank you for your feedback. We have expanded our discussion to encompass a broader consideration of the auditory and language pathways in speech processing. This includes an exploration of both hierarchical and parallel theories of acoustic-phonological mapping involving HG and STG, as well as the dynamic interactions between STG and higher-order brain regions during speech perception. We have also addressed the implications of our findings within this context (Page 23-24, Line 584-619). For more details, please refer to our response to Reviewer 2's Comment 2 below.

5) Add discussion of the possibility that the quantitative difference between effects on central auditory and language structures is due to the different ability to image each pathway.

Thank you for your valuable comment. We agree that the inherent challenges in imaging each pathway might indeed contribute to the observed differences in their structural effects. To address this concern, we have added an in-depth discussion on this point in the sections where we discuss the respective results of the auditory and language pathways (Page 22, Line 524-545). Specifically, we have included considerations on factors such as the severe distortion in diffusion images of the auditory pathway, particularly within the brainstem, the complex fiber crossing issue that is more pronounced within the auditory pathway, and the inherent differences in fiber numbers between the auditory and language pathways.

“Indeed, the quantitative differences observed between the effects on the central auditory and language structures may be influenced by the varying ability to accurately image each pathway with current neuroimaging technologies. The central auditory pathway, largely situated in the brainstem, faces severe distortion in diffusion images due to the inherent properties of the echo-planar imaging (EPI) sequence used. Despite distortion correction methods, it is challenging to entirely eliminate these effects. In contrast, the language pathways, primarily located in the cerebrum, have less distortion, allowing for more accurate imaging. Additionally, the auditory pathways in the brain (e.g., the auditory radiations) intersect with numerous cerebral fibre tracts, complicating accurate tractography. While the Constrained Spherical Deconvolution (CSD) modelling approach we employed is currently the most advanced method for addressing crossing fibres, it is not entirely exempt from these issues. The language pathways, on the other hand, intersect with fewer fibre tracts, facilitating more precise reconstruction. Lastly, the language pathways are more extensive and possess more fibres than the auditory pathways. From a statistical perspective, this enables more "sampling" when extracting average metrics such as fibre density, resulting in more stable outcomes. Therefore, the differences in our ability to visualize and quantify these pathways might have influenced the comparative effects observed. As neuroimaging technologies and methodologies continue to advance, we anticipate that our understanding of the differential effects of hearing loss on the central auditory and language structures will continue to evolve and improve.”

Additionally, we have included a summary of this discussion in the 'Limitations' section of our paper (Page 27, Line 654-658) to underscore its potential impact on our results. We hope that these additions sufficiently address your comment.

Reviewer #1 (Recommendations for the authors):Overall, the authors have done a beautiful job creating high-resolution maps of the auditory pathway in these children. This type of imaging is not my speciality and so I cannot comment on the methodology or analysis behind the imaging. However, I am an expert on pediatric hearing loss and so I evaluated this manuscript from that perspective.Detailed recommendations are below:1) There are some major deficiencies with respect to the clinical evaluation/presentation of the children in this study which makes interpretation of the results difficult. First of all, there is no diagnosis provided. The causes of pediatric hearing loss is extraordinarily complex with hundreds of genetic forms and environmental causes as well. Hearing loss is not a diagnosis but is instead a symptom of an underlying pathologic change to the peripheral and/or central auditory system. The authors do not provide a diagnosis for the 13 children with hearing loss and as such any conclusions are limited. For example, if some of the children have GJB2 hearing loss, I would expect differences compared to children with CDH23 hearing loss (which has been shown to have some central auditory effect). Similarly, variants in CHD7, are associated with CHARGE syndrome and are a common cause of cochlear nerve deficiency, with some presumed central effect. The authors should provide more clinical history on individual patients, including hearing testing, and should at the least have performed genetic testing for a full evaluation of hearing loss. Ideally, the authors would then separate their subjects into groups (i.e. those with GJB2 hearing loss, versus others). This would make their study more powerful.

We appreciate the reviewer's insightful comment regarding the complex etiologies of pediatric hearing loss, and the importance of understanding the specific diagnoses, including potential genetic causes, to fully interpret our results. We agree that the genetic basis of hearing loss could potentially influence the outcomes we have studied, and that this is a limitation of our current study. As stated in our response to the Essential Revision, we have now included more detailed clinical, auditory, and available genetic data for all patients in our study.

To further elucidate our study participants' audiological profiles, we have included a detailed table in the supplements. This comprehensive table delineates each patient's audiological history and current status, including newborn hearing screening results, initial diagnosis parameters, intervention details, and recent preoperative audiometric evaluations. We believe this enriched data set will provide valuable context for interpreting our study's results (see Supplementary file 1).

Notably, all children included in our study have been clinically characterized as having non-syndromic hearing loss (see Supplementary file 2). Comprehensive family histories have been obtained, with no reported congenital hearing loss or known genetic disorders within their immediate families. In addition, full-body examinations have been performed to exclude any syndromic forms of hearing loss. Therefore, while we appreciate the importance of genetic factors as pointed out by the reviewer, it is important to note that our cohort is composed of children clinically diagnosed with non-syndromic hearing loss. To enhance clarity, we have included this information in the first paragraph of the ‘Introduction’ (Page 2, Line 9-14).

“The etiology of these impairments is multifaceted, with genetic factors playing a significant role, alongside external contributors such as congenital infections (Marazita et al., 1993). Hearing loss is categorized as syndromic, with accompanying physical or laboratory findings, or non-syndromic, where hearing loss is the sole symptom and has a highly heterogeneous genetic basis (Korver et al., 2017). This article primarily focuses on non-syndromic sensorineural hearing loss.”

We have added the available genetic testing results for half of our patients (seven out of thirteen, see Supplementary file 2) and have included a discussion on the possible implications of these results for our study's findings (see Page 21, Line 469-490). However, it is not feasible for us to recall the remaining patients for this testing due to practical constraints. We acknowledge in the manuscript that the absence of genetic testing results for all patients limits our ability to draw definitive conclusions regarding the impact of specific genetic variants on our findings (see Page 27, Line 649-653).

(Page 21, Line 469-490) “Our findings revealed a stronger and more significant correlation between language pathway fibre metrics and peripheral nerve tissue density than with central auditory metrics. This might seem counterintuitive given the structural and functional connections between the central auditory pathway and the peripheral cochlear nerve. For patients with profound SNHL, who present normal cochlear nerves and inner ear structures, the primary pathology is often located at the cellular level within the cochlea. However, in non-syndromic profound SNHL patients exhibiting IEM&CND, more severe genetic abnormalities are typically suspected. In our study cohort, genetic testing results were acquired for seven participants, revealing mutations in GJB2 (two patients), OTOF (one patient), and MYO15A (one patient). These mutations are largely associated with peripheral auditory deficits, given these genes' critical roles in inner ear development and function (Kelsell et al., 1997; Stelma & Bhutta, 2014; Wang et al., 1998; Yasunaga et al., 1999). Consequently, we hypothesized that observed central auditory pathway alterations could be adaptive changes due to these peripheral deficits. Nevertheless, it remains uncertain whether the detected central auditory anomalies stem from genetic factors directly or whether they are the result of underdevelopment due to auditory deprivation following peripheral dysfunction. Our findings suggest a weak association between profound hearing loss-related impaired fixels in the central auditory pathway and cn.VIII tissue density, casting doubt on the former hypothesis. It is important to consider, however, the potential of certain gene mutations such as CDH23 or CHD7 – typically linked with syndromic forms of hearing loss – to directly affect central auditory pathways (Astuto et al., 2002; Zentner et al., 2010). Although our study primarily included non-syndromic hearing loss patients, these potential genetic influences highlight the complexities inherent in interpreting auditory function and call for further in-depth investigation.”

(Page 27, Line 649-653) “We also acknowledge some limitations to this study: (1) The genetic dataset in our study is incomplete and heterogeneous. While two patients underwent extensive testing of over 200 genes, five were tested for select common genes. For a thorough understanding of the genetic factors in auditory deficits, future studies should employ uniform and comprehensive genetic testing across participants;”

Subsequent adjustments to the methodological and Results sections have been made accordingly, which can be found in Page 28, Line 701-705 (Methods) and Page 5, Line 117-119 (Results).

We hope that these additions address the reviewer's concerns, and we thank them for prompting us to more thoroughly discuss these important issues.

2) This study does not include cochlear implant and/or auditory brainstem implant outcomes. Therefore, the authors cannot have any claims to determining or predicting outcomes with these devices until their correlate this finding with those results. This would be an excellent follow-up study and the authors should temper their statements on clinical applicability until this is completed.

We appreciate your suggestion to include post-implantation outcomes to strengthen our findings and their clinical applicability. As addressed in our response to the essential revisions, we have now incorporated these outcomes into our analysis (see Page 15-17). Additionally, we have introduced a section titled “Integrating pre-surgical evaluations for optimized auditory-language rehabilitation” in the ‘Discussion’ to elaborate on the implications and limitations of these results (see Page 23-25). This inclusion lends more substantiation to our assertions regarding the predictive value of the pre-operative MRI metrics for CI/ABI outcomes.

3) I assume that the CIs and ABIs were implanted after imaging? This is not specified anywhere that I could find.

Yes, both CT and MRI scans were conducted preoperatively during the evaluation phase for CI or ABI implantation. To enhance clarity, we have now restructured our 'Methods' section into five major subsections, namely: 'Study sample,' 'Pre-operative clinical and imaging evaluation', 'Post-operative outcomes assessment,' 'MRI data analysis,' and 'Statistical analysis.' The 'Clinical data' and 'CT, MRI acquisition, and quality assessment' aspects have been incorporated under the 'Pre-operative clinical and imaging evaluation' subsection.

4) The figure legends are confusing. Please indicate whether each image is an individual patient or an average and which group each image belongs to.

Thank you for your comment. We have made revisions to the legends in Figure 1 and Figure 2 to provide clearer information. We apologize for any confusion caused. If there are any other specific points that require clarification or if you have further suggestions, please let us know, and we will address them accordingly. Your feedback is greatly appreciated.

5) The graphs in Figure 3 are very very difficult to read. The authors should reformat these so that there is a white background and the bars are filled (or at least not use the current color scheme).

Thank you for your comment. We apologize for any difficulties in interpretation caused by the current color scheme and formatting. We have made the necessary adjustments. If you have any other specific recommendations or concerns, please let us know, and we will address them accordingly.

6) The authors should be careful about how they use the word 'deaf'. 'Deaf or hard of hearing' is an appropriate way to describe individuals with hearing loss. Severe-to-profound or profound are accurate ways to describe the severity of a hearing loss. The use of the word 'deaf' to describe profound hearing loss is generally discouraged as it is not scientific and may be confused by 'Deaf' (capital D) which describes members of the Deaf community.Please see above.

We appreciate this comment. The revised terms are now highlighted in green color throughout the manuscript. We consistently use 'profound hearing loss' to describe the patients in this study (hearing level over 95 dB at both sides before operation) and choose accurate descriptions when referencing the work of other authors.

Reviewer #2 (Recommendations for the authors):General comments– Although the paper tried to justify why focusing on IEM and CND is important, such decision seems still arbitrary to this reviewer. More discussion on what makes these hearing loss etiologies distinct from other etiologies would help.

Thank you for your valuable feedback. We understand the reviewer's concern regarding the justification for focusing on IEM&CND and the need to address it more comprehensively.

Emphasizing IEM & CND is crucial because it poses increased challenges in clinical decision-making compared to cases with normal inner ear structures, particularly in determining the suitable surgical intervention.

It is worth noting that the etiology of IEM&CND involves both genetic and environmental factors, similar to patients with normal inner ears. However, the genetic basis of non-syndromic IEM&CND is not extensively studied, which makes establishing clear genetic correlations challenging. Nonetheless, it is generally understood that the developmental arrest associated with IEM&CND occurs earlier compared to cases with normal inner ears.

To address these concerns, we have revised the second paragraph of the ‘Introduction’ (see Page 2, Line 15-43). This revised section elaborates on the challenges in clinical management posed by IEM&CND, provides an explanation for its complex etiology, and highlights the limitations of contemporary clinical imaging in diagnosing, evaluating prognosis, and guiding surgical strategies for profound SNHL patients, particularly those with IEM&CND.

“Inner ear malformations and cochlear nerve deficiencies (IEM&CND), identifiable with CT and MRI, contribute to 15-39% of pediatric sensorineural hearing loss (SNHL) cases (Li et al., 2011; Mafong et al., 2002). The rest is primarily due to cellular-level abnormalities. The etiology of IEM&CND is complex and largely unknown, yet these abnormalities suggest an earlier developmental arrest compared to cases with normal inner ear structures (Sennaroglu & Saatci, 2002). Addressing IEM&CND is vital as it presents specific challenges in clinical management. Cochlear implantation (CI) and auditory brainstem implantation (ABI) are currently the only solutions for profound SNHL, but the choice between CI and ABI presents a dilemma for many IEM&CND patients. CI directly stimulates the spiral ganglion cells, the first-order neurons of the auditory pathway, while ABI bypasses the cochlear nerve and stimulates the second-order auditory neurons in the cochlear nucleus when complex inner ear malformation or cochlear nerve aplasia makes CI inapplicable (Chen & Oghalai, 2016). Both CI and ABI are capable of providing hearing sensation and language development for children with severe-to-profound prelingual hearing loss, but postoperative outcomes vary among individuals. ABI recipients generally have poorer speech recognition performance and delayed and incomplete language development compared to CI recipients (Sennaroğlu et al., 2016). However, the prognosis of CI may not necessarily be better than that of ABI for certain IEMs, including common cavity, cochlear hypoplasia, incomplete partition-type I, and cochlear aperture abnormalities, as the presence of sufficient cochlear fibres required for CI success is uncertain (Freeman & Sennaroglu, 2018; Sennaroğlu & Bajin, 2017). This uncertainty is hard to address for two reasons: (1) assessing the cochlear nerve through visual inspection of MRI poses challenges, including subjective limitations, image quality, and difficulty distinguishing the cochlear nerve from the cranial nerve VIII (the cochleovestibular nerve) in cases of common cavity; and (2) the absence of the cochlear nerve structurally does not always indicate a lack of functional hearing (Thai-Van et al., 2000). Furthermore, distinguishing certain types of IEMs can be problematic and may impact surgical decision-making. For example, differentiating cochlear aplasia with a dilated vestibule (CADV) from a common cavity is challenging; CADV is a definitive indication for ABI, while a common cavity allows for either CI or ABI. These observations underscore the importance of a deep dive into IEM&CND, and highlight the limitations of contemporary clinical imaging in diagnosis, prognosis evaluation, and surgical guidance.”

Additionally, the penultimate paragraph of the introduction reinforces the inclusion of IEM&CND in our study (Page 3, Line 83-88).

“Furthermore, previous neuroimaging studies of profound congenital SNHL excluded children with IEM&CND. However, as mentioned earlier, these subjects comprise a significant proportion of patients with congenital hearing loss, and are more inclined to be faced with difficult surgical decisions and unsatisfactory post-implantation outcomes. Therefore, it is important to include and focus on children with IEM&CND when studying central adaptations associated with profound hearing loss.”

We hope that these modifications effectively justify the significance of focusing on IEM&CND and adequately address the concerns raised by the reviewer.

– The paper does not discuss how these results help with a better understanding of speech sound encoding and decoding in listeners with normal and impaired hearing. It is particularly important to discuss these findings in the context of mapping sound patterns to phonological units.– Also, there is a rich literature on hearing, language comprehension, and speech perception network from Gregory Hickok, David Poeppel, and Jonathan Peelle which are not discussed in this paper.

We appreciate your feedback regarding the need to contextualize our findings in the understanding of speech processing in listeners with normal and impaired hearing.

We opted to employ Angela D. Friederici's language model, given its explicit elucidation of the neuroanatomical fiber bundles associated with specific language functions. This aspect is particularly aligned with our study's emphasis on white matter structure. While Hickok and Poeppel's dual-stream model offers valuable insights, it does not directly associate the processing streams with specific neuroanatomical fiber bundles. However, it is noteworthy that there are considerable similarities between the two models, as both recognize the role of ventral streams in semantic processing and dorsal streams in sensorimotor integration. Additionally, Friederici's model includes a dorsal stream dedicated to syntactic processing. Our reason for basing our language pathway tracking on Friederici's model has been added in the Discussion section (Page 20, Line 437-439).

Further, we have expanded our discussion to elaborate on how our findings contribute to a better understanding of speech processing. We have included a discussion on both the hierarchical and parallel theories of acoustic-phonological mapping in HG and STG. Additionally, we have highlighted the significance of the dynamic interplay between STG and higher-level brain areas in speech perception, particularly under challenging conditions such as the use of auditory implants. We have discussed the implications of our findings in relation to these aspects. Please refer to Page 23-24, Line 584-619 for the revised content.

Title– The title needs to be more specific to the questions of this study. Using the general term "Peripheral Auditory Structure" could be misleading. Please revise to reflect both the specific types of hearing loss in this study and also the population that is children with profound hearing loss, but receive CI or ABI as interventions.

Thank you for your feedback. We agree that the term "Peripheral Auditory Structure" may not accurately convey the focus of our study. As a result, we have revised the title to "Impact of Inner Ear Malformations and Cochlear Nerve Deficiencies on the Development of Auditory-Language Network in Children with Profound Sensorineural Hearing Loss" to address this concern.

Regarding the decision not to include "receive CI or ABI" in the title, our intention was to avoid any potential ambiguity that could lead readers to assume that our study primarily focuses on post-implantation developmental changes in the auditory-language network. However, what we analyzed are the pre-implant MRI structures. We sincerely appreciate your valuable feedback, and we have taken your comments into account to enhance the clarity and accuracy of our study.

Additionally, we have updated the title of the fifth subsection in 'Results,' changing it from 'Peripheral auditory structure moderated the structural development of central pathways' to 'Peripheral nerve structure…' for enhanced clarity (Page 12, Line 200).

Abstract:– Cochlear implantation and auditory brainstem implantation can provide hearing sensation… it is more accurate to say … provide "partial" hearing sensation …

We agree with this comment and have added the word “partial”.

– "Previous attempts to locate subcortical auditory nuclei using fMRI responses to sounds are not applicable to deaf patients." Please add a reason(s) for why these previous attempts are not applicable.

We appreciate this comment and have added the reason as follows.

“Previous attempts to locate subcortical auditory nuclei using fMRI responses to sounds are not applicable to patients with profound hearing loss as no auditory brainstem responses can be detected in these individuals, making it impossible to capture corresponding blood oxygen signals in fMRI.”

– Line 3: lifelong "negative" consequences …

We agree with this comment and have added the word “negative”.

– Line 39: please check the in-text citations format

Thank you for your comment. We have followed the APA style, as suggested by *eLife*. In regard to the two references published in 2019 with different authors named "Wang," we used their initials to differentiate them. However, we have now updated the references to use the first and second authors for proper distinction (see Page 3, Line 57). If you have any specific requirements or further recommendations regarding this scenario, please let us know.

“(H. Wang et al., 2019; S. Wang, et al., 2019; Wu et al., 2016)” >> “(Wang, Liang, et al., 2019; Wang, Chen, et al., 2019; Wu et al., 2016)”

– Line 52: "Secondly, although earlier studies reported several altered fibre tracts related to language function, …" Please add references to support this statement.

Thank you for your comment. To address any potential misunderstanding, we have included the references again in Line 72-73, even though they were previously mentioned in the paragraph from Line 54-59.

– Line 83: mean [SD], age, 30.92 [6.115] months; 9 males, 4 females– Mean [SD] of age– You can remove 4 females as it is redundant– Please consider the same edit for NH– Line 85: Please indicate the confidence interval in your stats report

We agree with these two comments and have revised them in Page 5, Line 105-110.

“Twenty-three children aged under six years old including thirteen patients with bilateral profound congenital sensorineural hearing loss (mean [SD] of age, 30.92 [6.115] months; 9 males) and ten normal hearing volunteers (mean [SD] of age, 42.90 [4.270] months; 5 males) matched on age and gender were included (Mann-Whitney U test: p-value for age = 0.077, 95% confidence interval for the difference in age = [-32.0, 6.5] months; p-value for gender = 0.446, 95% confidence interval for the difference in male counts = [-1, 4]).”

Table 1– I recommend using the term gender instead of sex

We appreciate this comment and have revised “sex” to “gender” through the manuscript.

Line 137– FC, FD, and FDC are presented here for the first time in the manuscript. So, their extended version should be used here to avoid possible confusion.

We agree with this comment and have provided the extended version in Page 10, Line 167-168.

“In the central auditory pathway, FBA results demonstrated reduced fibre density (FD), fibre cross-section (FC), and fibre density and cross-section (FDC) in TB and decreased FC in LL in patients with profound SNHL.”

Caption of Figure 3: please report the actual p-values

Thank you. We have revised it in Figure 3 as well as in Figure 4.

– Line 139: ie. ◊ i.e.

Thank you. We have corrected this (Page 10, Line 170).

Figure 3– I wonder how the data and results would look if ABI and CI groups were represented separately.

Thank you for your comment. We recognize the merit in examining the differences between the ABI and CI groups in Figure 3. Although the primary objective of Figure 3 was to compare the hearing-impaired patients as a collective group against the normal control group, we have addressed your request by incorporating an additional figure (Figure 3—figure supplement 1), which compares the ABI, CI, and NH groups. Initially, we employed a one-way ANOVA to assess FD, FC, and FDC at the fixel-level with FWE correction, but no significant fixels were observed, possibly due to the small sample size. Subsequently, tract-level measures were used to illustrate the trend of differences among these three groups in Figure 3—figure supplement 1. We did not perform an ANOVA or post hoc tests at the tract-level to avoid the double dipping issue, especially since the fixel-level ANOVA did not yield significant results.

– Line 167-168: please elaborate on why such surgical decisions have limited value.

Thank you for the comment. We have elaborated on the challenges faced in making surgical decisions for patients with IEM&CND in a new paragraph in the ‘Introduction’ section, where we discuss the limitations of relying solely on visual inspection through CT and MRI (please refer to our response to your first comment). In light of this addition, we have decided to remove this sentence in the ‘Results’ section as the issue is now addressed in the ‘Introduction’.

– Line 193: Report the actual p–values

Thank you. We have reported the actual p-values through the section ‘Peripheral nerve structure moderated the structural development of central pathways’ to improve clarity (Page 12).

– Line 451: Provide more details on the nature of click stimuli including frequency.

Thank you for pointing out this omission. We used standard brief acoustic signals, also known as "clicks," for our study. These click stimuli have a duration of 100 microseconds and cover a broad frequency range from 100 Hz to 10,000 Hz. We have included these details in the ‘Methods’ section in Page 28, Line 695-698.

– Line 453: Clarify the dB HL used for NH decision (20 dB HL?).

Yes, we have added the standard (see Page 28, Line 682).

“The controls had normal hearing in both ears (click ABR threshold/PTA < 20 dB HL).”

– Line 468: If these measurements were not used in this study, it is not clear to this reviewer why they are reported. Please clarify.

Thank you for your observation. We acknowledge the lack of clarity in our initial manuscript. As indicated in our responses to the essential revisions and Reviewer 1's comment, we have now included post-implant behavioral outcome measures into our analysis (see Page 15-17 and Page 23-25). This update serves to validate the utility of our pre-operative MRI metrics for predicting post-implant behavioral outcomes.

– Line 479–481: please add inter–rater reliability to provide evidence on the acceptable agreement between raters.

Thank you for your suggestion. We have added the inter-rater reliability information to the relevant section (see Page 28, Line 713) of our manuscript to provide additional evidence for the reliability of our findings.

– Line 503–504: please report how many images were excluded in this process if any.

Thank you for your comment. It's important to clarify that no images were excluded during the quality control process. In one instance, an inappropriate head position was observed at the beginning of the scanning session, but it was promptly identified and corrected to ensure reliable data collection. As this issue was promptly resolved and did not impact the final dataset, it was not mentioned in the manuscript.

– Line 511: Does {plus minus}0.010 refer to standard deviation? If so, please indicate that in the paper.

Yes, we have clarified it (see Page 29, Line 753-756).

“The Cn.VIII was segmented on each axial section using ITK-SNAP (Yushkevich et al., 2006) by two independent raters. The inter-rater agreement was assessed using the DICE coefficient, yielding a value of 0.977±0.010 (mean± SD).”

– Table 1 and 2 in appendices: round correlation coefficients to two decimal places. More than two decimal places seem redundant.

We agree with this comment and have rounded correlation coefficients to two decimal places (now in Figure 4-source data 1).

Reviewer #3 (Recommendations for the authors):Recommendations:1. Enhance the reliability and replicability of research findings by increasing the sample size for both SNHL and NH children. The current sample size is insufficient for between-group comparisons and correlational analyses, particularly when examining sub-groups of SNHL children with only 6 or 7 subjects per group.

Thank you for your valuable comments. We wholeheartedly agree that a larger sample size can provide more robust and generalizable results. However, in the context of our current study, we are confronted with practical constraints, including the unique characteristics of our target population and limited available resources. Expanding the participant pool requires specific procedures such as patient recruitment, informed consent, data collection, and data processing. Given the scope of our current study and resources, it is challenging to significantly expand our sample size at this stage. Despite this limitation, we believe our findings are valuable, shedding light on an area with limited prior research. We have exercised rigorous statistical controls and analyses to maximize the validity and reliability of our findings within the confines of our sample size. We recognize the necessity for further research with larger, more diverse cohorts, and have acknowledged this need in the limitations section of our paper. Thank you once again for your constructive suggestions.

2. Incorporate post-surgical behavioral outcome measures into the analysis to assess the value of pre-surgical neuroimaging and neural analyses. This will help elucidate the relationship between pre-surgical neuroimaging and post-surgical behavioral benefits, informing clinical decision-making.

We appreciate your suggestion to incorporate post-surgical behavioral outcome measures into our analysis. As we mentioned in our responses to previous comments, we have incorporated these measures into our analysis (see Page 15-17 and Page 23-25). The correlation between pre-operative MRI metrics and post-implant behavioral outcomes further supports the potential utility of our imaging markers in informing clinical decision-making.

3. Reevaluate data analyses for potential double dipping issues, specifically in lines 153-155 and Figure 3 C&D. The impaired pixels identified by comparing SNHL and NH groups are re-analyzed for the same comparison, potentially inflating effect size by only examining the ROI/pixels derived from the same comparison.

Thank you for your comment. We recognize that there is indeed a risk of inflating the effect size by re-analyzing the impaired pixels identified from the same comparison.

In our study, we opted for ROI-based analysis to visually demonstrate the fiber structure of each segment in the auditory pathway, as it offers a clear and intuitive representation. We used FWE correction as a statistical control to mitigate the likelihood of false positives in the ROI analysis. However, we acknowledge that FWE correction does not directly address the potential double dipping issue.

We have acknowledged this limitation and included a note in the Results section of our paper (Page 10, Line 186-189). Your feedback has prompted us to provide a more explicit explanation of the limitations and considerations associated with our analysis methods. We sincerely appreciate your valuable input.

“It is worth noting, however, that in our tract-of-interest analysis, there is the potential for double dipping as we re-analysed the impaired pixels identified from the same comparison, which could have impacted the reported effect size.”

4. To better understand age effects, conduct the analyses presented in Figure 4 for NH children, which will serve as a baseline to demonstrate how different types of SNHL modulate the development of auditory and language pathways in comparison to normal development.

Thank you for your comment and suggestion. We appreciate your suggestion about studying the relationship between peripheral nerve development and central auditory and language pathways in normal condition.

However, we would like to mention that in our study, the normal hearing control group did not undergo scanning with the CISS sequence, which is necessary for measuring peripheral nerve indicators. Consequently, we currently lack the data required for a direct comparison of the development of auditory and language pathways between NH children and those with different types of sensorineural hearing loss.

We have acknowledged this limitation in our study and have discussed it in the relevant section of discussion (Page 22, Line 541-545). We sincerely appreciate your feedback, and it will undoubtedly guide our future research efforts.

“Another limitation is that the normal hearing control group did not undergo scanning with the CISS sequence, precluding the establishment of a baseline for comparing the development of auditory and language pathways in SNHL to normal development, which requires future research to include a baseline assessment.”

5. Ensure that modulation effects displayed in Figure 4 are controlled for confounding variables, such as residual hearing, sex, gestational weeks, birth weight, etc. to improve the validity of the results.

Thank you for your valuable comment. We acknowledge that these variables could potentially impact the validity of the results. We acknowledge the importance of considering residual hearing as a potential factor; however, it is worth noting that all patients in our study exhibited click ABR and/or pure tone thresholds (at 0.5k, 1k, 2k, and 4k Hz) that were over 95 dB HL bilaterally, indicating a profound degree of hearing loss. In fact, the pure tone audiometry results showed thresholds over 95 dB HL at each frequency of 0.5k, 1k, 2k, and 4k Hz for all patients who took the exam. Given this, we did not include residual hearing as a covariate in our analysis. However, we have re-calculated the moderation analyses while controlling for other variables including gender, gestational weeks, and birth weights. The results of these analyses can be found in Figure 4-source data 2, and are described in the relevant sections under ‘Results’ (Page 12, Line 239-241) and ‘Discussion’ (Page 22, Line 517-519). The moderation effect became non-significant after controlling these variables, potentially due to a relatively small sample size in comparison to the number of control variables. We appreciate your insightful suggestion, as it has urged us to interpret the results with greater caution.

We hope that the revisions and our responses adequately address the reviewers' comments and meet the standards of *eLife*. We are open to further suggestions and look forward to your response.